# Contrastive Representation Regularization for Vision-Language-Action Models

**Taeyoung Kim** [1 2]    **Jimin Lee** [1]    **Myungkyu Koo** [1 2]    **Dongyoung Kim** [1 2]    **Kyungmin Lee** [1]
**Changyeon Kim** [1]    **Younggyo Seo** [3 †]    **Jinwoo Shin** [1 2 †]

## Abstract

Vision-Language-Action (VLA) models have shown strong capabilities in robot manipulation by leveraging rich representations from pretrained Vision-Language Models (VLMs). However, their representations arguably remain suboptimal, lacking sensitivity to robotic signals such as control actions and proprioceptive information. To address the issue, we introduce *Robot State-aware Contrastive Loss (RS-CL)*, a simple and effective representation regularization for VLA models, designed to bridge the gap between VLM representations and robotic signals. In particular, RS-CL aligns the representations more closely with the robot's proprioceptive states by using relative distances between the states as soft supervision. Complementing the original action prediction objective, RS-CL enhances control-relevant representation learning, while being lightweight and fully compatible with standard VLA training pipelines. Our empirical results demonstrate that RS-CL substantially improves the performance of state-of-the-art VLA models; it pushes the prior art to 69.7%, achieving the state-of-the-art performance on the RoboCasa-Kitchen benchmark, and boosts success rates from 45.0% to 58.3% on challenging real-robot manipulation tasks.

## 1. Introduction

Vision-Language-Action (VLA; Zitkovich et al. 2023) models have emerged as a powerful framework for robot manipulation, leveraging pre-trained Vision-Language Models (VLM; Liu et al. 2023b) to provide rich visual and semantic grounding for control policies. Among the state-of-the-art VLA models, the common design is to employ a generative action decoder conditioned on VLM-derived representa-

tions (Black et al., 2025b; Bjorck et al., 2025). These decoders are trained with an action prediction loss, supervised by the ground-truth sequence of actions.

Prior studies have shown that training the backbone VLM alongside the action decoder is essential to the action prediction performance of VLA models. This is because VLMs are typically trained on large-scale visual instruction datasets, but have not been explicitly exposed to robotic modalities, such as low-level control actions and proprioceptive information. Consequently, training VLA models conditioned on frozen VLM representations leads to suboptimal performance, as the VLM lacks the capability to capture robotic signals (Driess et al., 2025).

Many recent works have proposed methods to train the VLM backbone in VLA models to tackle this issue. A widely adopted strategy is to directly update the VLM via gradients from the action prediction objective (Black et al., 2025b; Bjorck et al., 2025). Beyond this, several works introduce auxiliary objectives, such as jointly training the VLM backbone with curated instruction datasets (Yang et al., 2025), or blocking gradients from the action decoder, and instead learning to generate intermediate subtasks and discretized actions (Driess et al., 2025). Another line of work further trains the VLM on embodied reasoning or spatial grounding tasks using robotics datasets (Cao et al., 2025; Luo et al., 2025; Azzolini et al., 2025; NVIDIA GEAR, 2025), or autoregressively predicts discretized actions (Kim et al., 2025; Black et al., 2025a) before fine-tuning them for continuous action prediction. While these approaches help bridge the gap between general-purpose VLM representations and the demands of action prediction, they often require additional training stages or carefully curated datasets.

In contrast, we aim to directly refine VLM representations to better serve action prediction, while remaining efficient and seamlessly compatible with the existing VLA training pipelines. In particular, we focus on contrastive learning, as it provides a principled way to refine representations by defining similar and dissimilar pairs, effectively structuring the embedding space. The specific choice of pair construction determines what the embeddings should capture, ranging from semantic relations between modalities (Radford et al., 2021) to temporal dynamics and policy-relevant

---

†Equal advising. [1]KAIST [2]RLWRLD [3]UC Berkeley. Correspondence to: Taeyoung Kim <taeyoungkim85@kaist.ac.kr>.

*Proceedings of the $43^{rd}$ International Conference on Machine Learning*, Seoul, South Korea. PMLR 306, 2026. Copyright 2026 by the author(s).

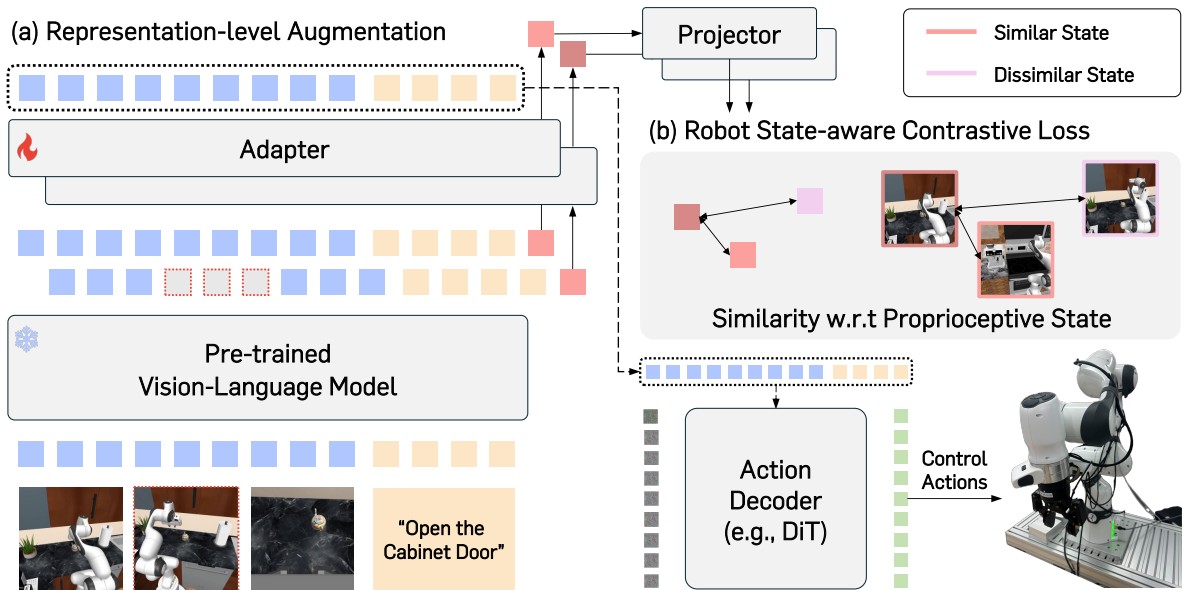

*Figure 1.* **Overview.** We extend the standard VLA model training framework with a contrastive regularization path. Embeddings from the pre-trained VLM are augmented by the *view cutoff* operation applied on the feature slice corresponding to a randomly selected observation view, and are optimized with our *Robot State-aware Contrastive Loss* to attract samples with similar proprioceptive states.

representations (Sermanet et al., 2018; Nair et al., 2022; Ma et al., 2023). Inspired by this perspective, we introduce a contrastive objective that explicitly guides the representations to capture robotic signals, in particular the robot's proprioceptive states. By jointly optimizing the VLM representation with the standard action prediction loss, we learn representations that are not only semantically rich but also deeply grounded in the robot's physical state, leading to accurate action prediction.

**Contribution.** We propose *Robot State-aware Contrastive Loss (RS-CL)*, a representation regularization that leverages continuous proprioceptive distances as soft contrastive supervision to reshape VLM representations for accurate action prediction. Unlike conventional contrastive losses, RS-CL assigns pairwise weights based on proprioceptive state distances, guiding representations to better reflect control-relevant structure. In addition, we propose a representation-level augmentation for VLA models, termed *view cutoff*. This augmentation constructs alternative embeddings by masking out the feature corresponding to a randomly selected observation view. By operating at the representation level and minimizing additional forward passes through the pre-trained VLM, RS-CL enables single-stage, end-to-end representation learning for VLA models, remaining fully compatible with existing training pipelines.

We extensively evaluate the effectiveness of RS-CL under multitask manipulation benchmarks such as RoboCasa-Kitchen (Nasiriany et al., 2024) and LIBERO (Liu et al., 2023a). RS-CL achieves state-of-the-art performance on the RoboCasa-Kitchen benchmark with a success rate of

69.7%, surpassing all strong baselines. Notably, RS-CL yields larger gains on pick-and-place tasks, improving success rates from 30.3% to 41.5% (+11.2%), which require precise positioning during grasping and placing. Finally, we show that RS-CL is applicable to real-robot hardware experiments, showing improvement from 45.0% to 58.3% (+13.3%) on challenging manipulation tasks.

In summary, our contributions are as follows:

- We introduce *Robot State-aware Contrastive Loss (RS-CL)*, a novel objective for VLA models that explicitly aligns VLM representations with proprioceptive states.

- We design RS-CL to operate directly at the representation alongside the original action prediction objective. Therefore, RS-CL remains lightweight and compatible with the existing training pipeline.

- We validate RS-CL across diverse training scenarios on simulation benchmarks and real-world experiments, yielding improvements over the state-of-the-art VLA models.

## 2. Method

In this section, we introduce *Robot State-aware Contrastive Loss (RS-CL)*, which enhances the action prediction capability of VLA models by guiding the representation to capture low-level robotic signals, particularly the proprioceptive states. We describe the VLA training framework in Sec. 2.1 and present our proposed method, RS-CL, in Sec. 2.2. An overview of our method is shown in Fig. 1.

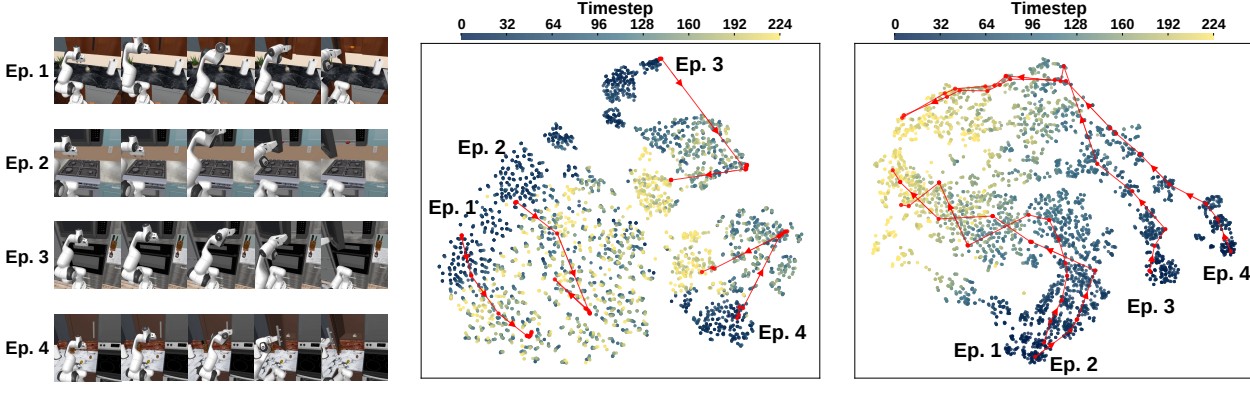

*(a)* Trajectory examples of identical task.  *(b)* Pre-trained VLM representations.  *(c)* RS-CL aligned representations.

*Figure 2.* **Training VLM representations towards action prediction. (a)** We visualize VLM embeddings of robot episodes performing the same task "Open the microwave / cabinet door" across different scenes in RoboCasa-Kitchen. **(b)** Pre-trained VLM representations are dominated by the visual appearance (*e.g.*, scene layout, large surrounding objects). **(c)** RS-CL guides embeddings to align with the robot's proprioceptive states, yielding representations that capture common robotic signals (*e.g.*, the robot's current pose, next control action) across environments, therefore aligning all episodes by the task progress.

## 2.1. Vision-Language-Action model

VLA models are trained to predict the next action chunk $\mathbf{A}_t = [\mathbf{a}_t, \mathbf{a}_{t+1}, \ldots, \mathbf{a}_{t+H}]$ of horizon $H$ at current timestep $t$, from a set of observation images from $V$ different views $\mathbf{O}_t^V = \{\mathbf{o}_t^1, \mathbf{o}_t^2, \ldots, \mathbf{o}_t^V\}$, a task instruction $\mathbf{c}$, and the robot's proprioceptive state $\mathbf{q}$. A standard framework for VLA models (Black et al., 2025b; Bjorck et al., 2025) encodes multimodal inputs $[\mathbf{O}_t^V, \mathbf{c}]$ using a pre-trained VLM into a hidden representation, and pass it to the action decoder. In practice, we train a lightweight adapter module $f_\phi$ upon the VLM and freeze the VLM, following NVIDIA GEAR (2025). $f_\phi$ processes the output of the VLM as $\mathbf{h} = f_\phi\big(\text{VLM}(\mathbf{O}_t^V, \mathbf{c})\big) \in \mathbb{R}^{N \times d_{\text{model}}}$, where $N$ is the number of input tokens for the VLM and $d_{\text{model}}$ is the size of the hidden dimension.

An action decoder $D_\theta$ generates $\mathbf{A}_t$ conditioned on $\mathbf{h}$ with the current robot state $\mathbf{q}$. Similar to prior works (Black et al., 2025b; Bjorck et al., 2025), we adopt the DiT (Peebles & Xie, 2023) architecture for the $D_\theta$ and train with the flow-matching objective (Lipman et al., 2023; Liu, 2022):

$$\mathcal{L}_{\text{FM}}(\theta, \phi) = \mathbb{E}_s\Big[\|D_\theta(\mathbf{h}, \mathbf{A}_t^s, \mathbf{q}) - (\epsilon - \mathbf{A}_t)\|_2^2\Big], \quad (1)$$

where $\mathbf{A}_t^s = s\mathbf{A}_t + (1-s)\epsilon$ is an interpolated action chunk at the flow-matching timestep $s \in [0, 1]$ sampled from a prior distribution $p(s)$. After training, $D_\theta$ generates $\mathbf{A}_t$ through an iterative denoising process starting from a random Gaussian noise $\epsilon \sim \mathcal{N}(\mathbf{0}, \mathbf{I})$.

## 2.2. Robot State-aware Contrastive Loss

While VLMs acquire rich semantic representations from Internet-scale vision–language data, they lack exposure to robotic modalities such as low-level control actions and proprioceptive states. As a result, their embeddings are often dominated by the visual appearance and fail to capture signals relevant to robot control. This misalignment is evident when visualizing VLM embeddings of robot trajectories for the same manipulation task (*e.g.*, Open the microwave / cabinet door) across different environments in RoboCasa-Kitchen (see Fig. 2a). We observe that the embeddings cluster primarily by visual cues, such as large objects or background textures (see Fig. 2b), rather than control-relevant factors like the robot's current pose or the next action needed to complete the task. A more detailed visualization and analysis are provided in Appendix E.1.

This misalignment motivates our central hypothesis: explicitly aligning VLM representations with the robot's physical state will improve the action prediction performance of VLA models. Based on this hypothesis, we introduce *Robot State-aware Contrastive Loss (RS-CL)*, an auxiliary objective for VLA models that regularizes the VLM representation space using supervision from the robot's proprioceptive states. Our key idea is a contrastive loss that leverages the distances between proprioceptive states to assign soft weights to similarity scores, which effectively guides the representation space toward alignment with control-relevant robotic signals. As an auxiliary objective, RS-CL complements the action prediction loss, enabling the entire model to be trained end-to-end in a single stage. Concretely, RS-CL consists of three key components: (i) a *learnable summarization token* that amortizes long VLM output embeddings, (ii) a *weighting scheme* for robot state supervision via contrastive learning, and (iii) a *representation-level augmentation* strategy that enables lightweight and stable representation learning.

**Amortizing VLM embeddings for representation learning.** Applying contrastive learning on the full sequence of VLM embeddings $\mathbf{h} \in \mathbb{R}^{N \times d_{\text{model}}}$ is impractical as the sequence length $N$ is typically large, leading to high computational cost and diluted learning signals. To address this, we introduce a *learnable summarization token* $\mathbf{u} \in \mathbb{R}^{1 \times d_{\text{model}}}$ to produce a compact representative embedding of the sequence. Specifically, $\mathbf{u}$ is appended to the VLM output and processed by the adapter $f_\phi$:

$$[\mathbf{h}, \mathbf{w}] = f_\phi\big(\text{VLM}(\mathbf{O}_t^V, \mathbf{c}) \oplus \mathbf{u}\big), \quad (2)$$

where $\mathbf{w}$ denotes the output of the summarization token and $\oplus$ denotes concatenation along the sequence dimension. Finally, $\mathbf{w}$ is projected by a lightweight projector $g_\psi$ into $\mathbf{z} = g_\psi(\mathbf{w})$, providing a compact summary for contrastive learning (Chen et al., 2020), while the original embedding $\mathbf{h}$ serves as the conditioning input to the action decoder.

**Incorporating robot states into contrastive learning.** To effectively restructure the VLM representation space to capture robotic signals, we introduce a supervised contrastive learning objective assigned with *soft weights* (Khosla et al., 2020; Suresh & Ong, 2021), that incorporate the distance between proprioceptive states. Conceptually, embeddings associated with similar proprioceptive states receive higher weights, are pulled closer in the representation space. We consider InfoNCE (Oord et al., 2018) for the contrastive loss, which is widely used in practice (Laskin et al., 2020; Nair et al., 2022; Ma et al., 2023). Formally, our *Robot State-aware Contrastive Loss (RS-CL)* is defined as a weighted variant of the InfoNCE loss:

$$\mathcal{L}_{\text{RS-CL}}(\phi, \psi) = -\sum_{i,j=1}^{B} w_{ij} \log \frac{e^{\text{sim}(\mathbf{z}_i, \tilde{\mathbf{z}}_j)/\tau}}{\sum_{k=1}^{B} e^{\text{sim}(\mathbf{z}_i, \tilde{\mathbf{z}}_k)/\tau}}, \quad (3)$$

where $\{\tilde{\mathbf{z}}_j\}_{j=1}^B$ denotes the augmented batch corresponding to $\{\mathbf{z}_i\}_{i=1}^B$, sim denotes the cosine similarity, and $\tau > 0$ is a temperature that controls the sharpness of similarity. The soft weights $w_{ij}$ are computed from the relative distance between proprioceptive states $\mathbf{q}_i, \mathbf{q}_j$. In practice, we use the Euclidean distance and formulate $w_{ij}$ as follows:

$$w_{ij} = \frac{e^{-\|\mathbf{q}_i - \mathbf{q}_j\|_2/\beta}}{\sum_{k=1}^{B} e^{-\|\mathbf{q}_i - \mathbf{q}_k\|_2/\beta}}, \quad (4)$$

where $\beta > 0$ is a temperature that controls the sharpness of the mapping from distance to weight. The complete training objective integrates the proposed RS-CL with the action prediction objective, implemented as the flow-matching loss in Eq. 1:

$$\mathcal{L} = \mathcal{L}_{\text{FM}} + \lambda \mathcal{L}_{\text{RS-CL}}, \quad (5)$$

where $\lambda > 0$ is a weighting coefficient that balances the contribution of RS-CL, and we jointly optimize $\theta$, $\phi$, and $\psi$.

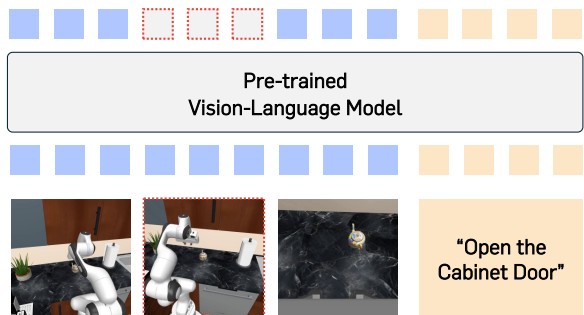

*Figure 3.* **Representation-level augmentation for contrastive learning.** *View cutoff* randomly masks the embedding slice associated with one observation viewpoint, forming contrastive pairs at the representation level.

**Representation augmentation for contrastive pairs.** The primary goal of our augmentation strategy is to generate diverse contrastive pairs while preserving the semantics tied to the robot's proprioceptive states. In line with this goal, we exploit the property that VLA models commonly process observations of the same scene from multiple views, and propose *view cutoff* (see Fig. 3), a simple representation-level augmentation inspired by cutoff (Shen et al., 2020). View cutoff randomly selects a single view index $i \in \{1, \ldots, V\}$ and masks out the corresponding feature slice from the VLM output $\text{VLM}(\mathbf{O}_t^V, \mathbf{c})$. Unlike data-level augmentations requiring additional forward passes through the VLM for each augmented batch, view cutoff operates at the representation level, obtaining alternative representations with minimal overhead. As a result, only the lightweight adapter $f_\phi$ and projector $g_\psi$ are required to process the augmented variants, making the method substantially more efficient, yet still providing diverse pairs for contrastive learning.

## 3. Experiments

In this section, we evaluate the effectiveness of RS-CL across diverse training scenarios. In Section 3.1, we examine its impact on a pre-trained state-of-the-art VLA model across multitask manipulation benchmarks: RoboCasa-Kitchen (Nasiriany et al., 2024) and LIBERO (Liu et al., 2023a), and further demonstrate its applicability to real-world tasks using a 7-DoF manipulator. In Section 3.2, we further validate RS-CL when training VLAs from scratch on top of various pre-trained VLM backbones.

**Implementation and training details.** We adopt GR00T N1.5 (NVIDIA GEAR, 2025) as our baseline VLA framework and unless otherwise specified, we follow the training and inference settings of the original implementation. For the regularization path, a projection head $g_\psi$ is implemented as a 2-layer MLP with hidden dimension 2048 and projection dimension 128. The weighting coefficient $\lambda$ for $\mathcal{L}_{\text{RS-CL}}$ is initialized to 1.0 and decayed to 0 using a cosine schedule, such that representation refinement is emphasized in

*Table 1.* **RoboCasa-Kitchen benchmark success rate (%).** Results include the performance of representative VLA methods trained with 300 demonstrations. Results of $\pi_0$, $\pi_0$-FAST, and GR00T N1.5 are reproduced while others are from the original work. Best results are highlighted in **bold**.

| Method | Avg. |
|---|---|
| GR00T N1 (Bjorck et al., 2025) | 49.6 |
| GR00T N1 + DreamGen (Jang et al., 2025) | 57.6 |
| GR00T N1 + DUST (Won et al., 2025) | 58.5 |
| $\pi_0$ (Black et al., 2025b) | 62.5 |
| $\pi_0$-FAST (Pertsch et al., 2025) | 63.6 |
| GR00T-N1.5 (NVIDIA GEAR, 2025) | 65.7 |
| Video Policy (Liang et al., 2025) | 66.0 |
| FLARE (Zheng et al., 2025) | 66.4 |
| GR00T N1.5 + HAMLET (Koo et al., 2026) | 66.4 |
| **GR00T N1.5 + RS-CL (ours)** | **69.7** |

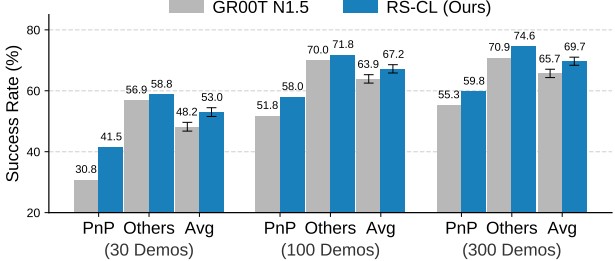

*Figure 4.* **RoboCasa-Kitchen benchmark success rate (%) by number of training demonstrations.** We compare RS-CL (blue) with the baseline GR00T N1.5 (gray) across varying training dataset sizes. RS-CL consistently outperforms across all settings, and notably demonstrates significant performance gains in the low-data regime (30 demos) for pick-and-place tasks.

early training while accurate action prediction becomes the main focus later. For proprioceptive states, we primarily use the end-effector position ($x$, $y$, $z$, 6D rotation, and gripper state), min-max normalized to $[-1, 1]$ per dimension. In the real-robot experiments, we additionally explore the use of absolute joint positions of the manipulator to examine variations in proprioceptive configurations.

### 3.1. Fine-tuning Experiments

We first evaluate RS-CL in a fine-tuning scenario, where it is integrated into a state-of-the-art pre-trained VLA model. This setup tests whether RS-CL can yield additional gains on model weights that are already optimized for action prediction in large-scale, demonstrating the ability to further enhance strong pre-trained policies at a target environment.

**Simulation experiment setup.** We evaluate RS-CL on two widely used multitask simulation benchmarks, RoboCasa-Kitchen and LIBERO, comparing against representative VLA baselines. RoboCasa-Kitchen consists of 24 atomic manipulation tasks in a simulated kitchen environment with three camera views (2 exterior, 1 wrist camera). We evaluate RS-CL under varying numbers of training demonstrations

*Table 2.* **LIBERO benchmark success rate (%).** Results include the performance of representative VLA methods. Results of GR00T N1 and GR00T N1.5 are reproduced, where the performance of $\pi_0$-FAST, $\pi_0$ are from the original work. Best results are highlighted in **bold**.

| Method | Spatial | Object | Goal | Long | Avg. |
|---|---|---|---|---|---|
| $\pi_0$-FAST | 96.4 | 96.8 | 88.6 | 60.2 | 85.5 |
| $\pi_0$ | 96.4 | 98.8 | 95.8 | 85.2 | 94.1 |
| GR00T N1 | 95.6 | 97.6 | 94.2 | 89.6 | 94.3 |
| GR00T N1.5 | 98.2 | **99.4** | 97.2 | 87.8 | 95.7 |
| + RS-CL (ours) | **98.4** | 98.6 | **98.2** | **90.4** | **96.4** |

*Table 3.* **Comparison to time-contrastive networks (TCN; Sermanet et al. 2018).** Results report the average success rate (%) and FLOPs per sample for a single forward pass during training on RoboCasa-Kitchen with 30 demonstrations. Best results are highlighted in **bold**, and $\uparrow$ ($\downarrow$) indicates higher (lower) is better.

| Method | SR (%, $\uparrow$) | FLOPs ($\times 10^{12}$, $\downarrow$) |
|---|---|---|
| Baseline | 48.2 | 2.58 |
| Multi-view TCN | 50.0 | 7.53 |
| Single-view TCN | 50.3 | 7.53 |
| **RS-CL** | **53.0** | **2.91** |

(30, 100, and 300) using the publicly available dataset generated by MimicGen (Mandlekar et al., 2023). LIBERO comprises four multitask suites with two camera views (1 exterior, 1 wrist camera). For LIBERO, we use the filtered dataset released by Kim et al. (2024) and jointly train all four task suites (see Appendix B for details).

**Simulation experiment results.** Table 1 and Fig. 4 summarize the performance of RS-CL on the RoboCasa-Kitchen benchmark. RS-CL achieves state-of-the-art performance among all VLA baselines and consistently outperforms the original GR00T N1.5 fine-tuning framework across all dataset sizes. In particular, RS-CL yields substantial gains on pick-and-place tasks, with success rates increasing from 30.3% to 41.5% (+11.2%). We attribute these gains to RS-CL's ability to produce more accurate actions during execution, which is especially important for pick-and-place tasks that require precise positioning during both grasping and placing. RS-CL also improves performance on LIBERO (Table 2), confirming its robustness across benchmarks.

To evaluate the effectiveness and efficiency of our specific design choice, we compare RS-CL with time-contrastive networks (TCN; Sermanet et al. 2018), a representative representation learning method in robotics that learns from explicitly comparing representations across neighboring timesteps. We implement TCN as an auxiliary objective on top of GR00T N1.5, considering both multi-view and single-view variants (see Appendix B.3 for details). Table 3 shows that both TCN variants provide only modest improvements over the baseline, while incurring a substantial increase in computational cost. This overhead arises

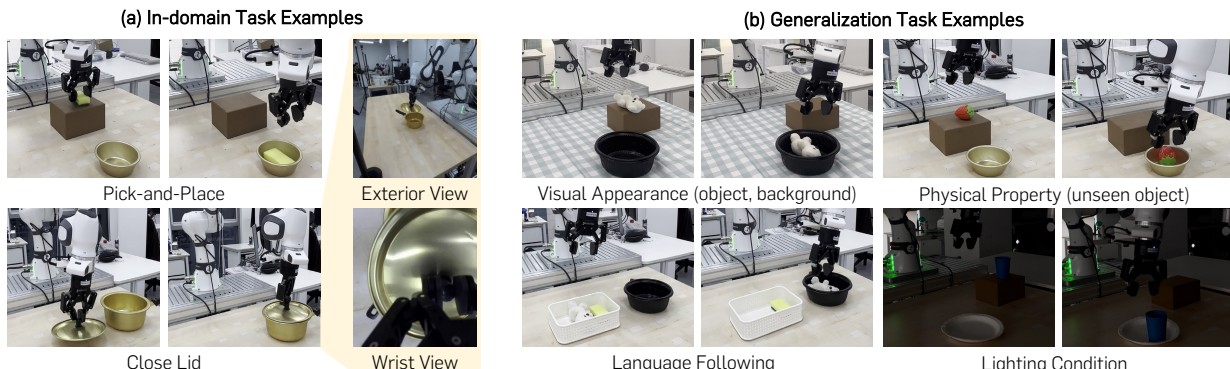

*Figure 5.* **Real-robot experiment task examples.** We validate RS-CL on a Franka Research 3 arm with a Robotiq 2F-85 gripper, utilizing two camera views. **(a)** We conduct pick-and-place tasks across diverse objects and environments, and a close lid task. **(b)** We evaluate generalization abilities under variations in visual appearance, physical properties, language following, and lighting conditions.

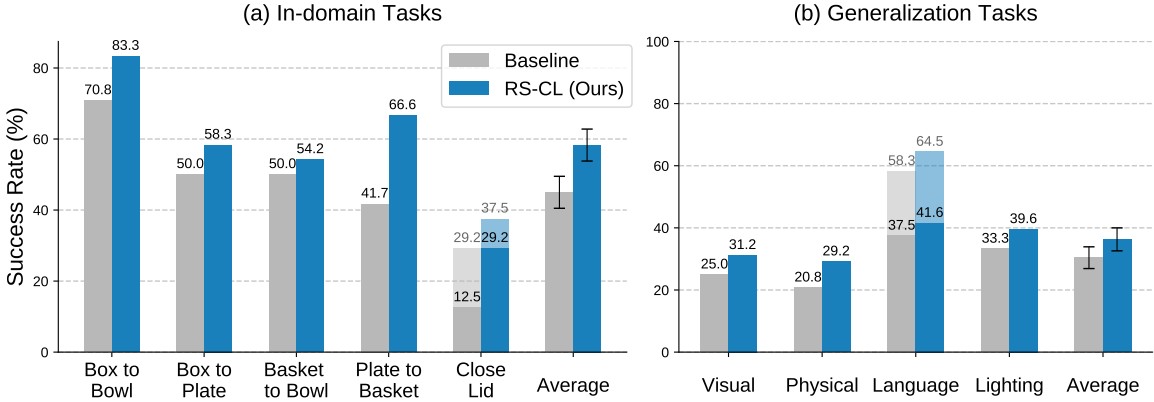

*Figure 6.* **Real-robot experiment success rate (%).** Results on **(a)** in-domain tasks (4 pick-and-place and 1 close-lid task), and **(b)** generalization tasks (visual, physical, lighting variation and language following). For the in-domain close lid and language following tasks, we report both partial success (*e.g.*, successful pickup, language following; transparent bars) and full success (solid bars).

from the need for temporally structured data mining and additional VLM forward passes to construct positive and negative pairs. In contrast, RS-CL achieves a higher success rate with only a modest increase in FLOPs. This efficiency is enabled by the proposed *view-cutoff* augmentation, which operates entirely at the representation level after a single VLM forward pass. Overall, RS-CL serves as an effective yet lightweight regularizer for single-stage, end-to-end VLA training, strengthening conditioning representations without significant additional computational overhead.

**Real-robot experiment setup.** To further assess whether RS-CL leads to more precise actions in task execution, we design our real-robot experiments primarily around pick-and-place tasks, which require accurate positioning during grasping and placing. We also introduce a challenging close lid task; the lid has a small handle that is more difficult to grasp than other objects, and once grasped, the wrist camera view becomes partially occluded, making placement harder (see Fig. 5a). We evaluate each method on both in-domain tasks and generalization tasks in Fig. 5. We collect and train each method with teleoperated expert demonstrations

of 4 pick-and-place tasks across diverse objects (teddy bear, sponge, cup, cube) and environments (box, bowl, plate, basket), as well as the close lid task, utilizing two camera views (exterior and wrist). For pick-and-place tasks, we use end-effector position as the proprioceptive state, while for the close lid task, we use absolute joint positions of the manipulator; see Appendix C for details.

**Real-robot experiment results.** RS-CL consistently improves performance across real-robot tasks (see Fig. 6a). In particular, for the close-lid task, RS-CL brings improvements not only in partial success (*i.e.*, lifting the lid) but also larger gains in complete success (*i.e.*, accurately closing the pot) even under occluded viewpoints (see Fig. 5a, bottom-right). We attribute this effect to two factors: (i) proprioceptive supervision enables more accurate positioning, and (ii) the proposed *view cutoff* augmentation promotes view-invariant representations, thereby improving robustness to partial occlusion. Finally, our generalization experiments show that RS-CL not only maintains but leads to stronger generalization performance of VLAs across visual, physical, lighting shifts, and language grounding (see Fig. 6b).

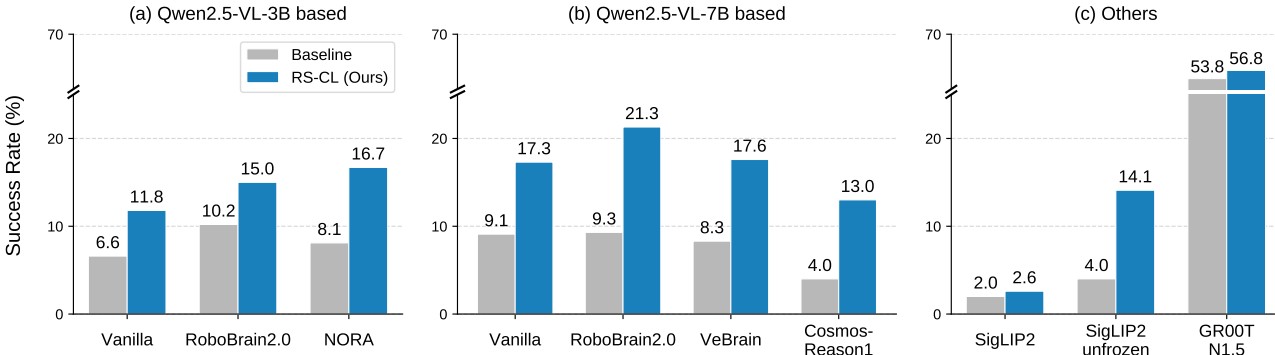

*Figure 7.* **From-scratch experiments across multiple backbone VLMs.** Success rates (%) on RoboCasa-Kitchen for VLA models trained from various backbones, where Vanilla denotes Qwen2.5-VL. Results show the effects of RS-CL on backbones further trained with robotics data, based on **(a)** Qwen2.5-VL-3B, **(b)** 7B, and **(c)** SigLIP2 and GR00T N1.5, covering diverse backbone and training capacities.

## 3.2. From-Scratch Experiments

In this section, we evaluate the impact of RS-CL in a from-scratch training scenario, where we train a VLA model directly on top of general-purpose pre-trained VLM backbones of Qwen2.5-VL (Bai et al., 2025) and SigLIP2 (Tschannen et al., 2025). This setup directly reflects our motivation that pre-trained VLM representations lack sensitivity to robotic signals, and enables us to validate whether explicitly aligning them to proprioceptive information yields performance gains. Furthermore, we compare RS-CL against baselines obtained by further training general-purpose VLMs on robotics datasets.

**Setup.** We adopt RoboCasa-Kitchen as our primary benchmark, using 300 demonstrations for training. We attach a randomly initialized action decoder to pre-trained VLMs with a lightweight adapter module $f_\phi$. The VLM backbone is frozen and only the adapter is trained to refine conditioning representations, except for SigLIP2, where we additionally try unfreezing the backbone to study the effect of RS-CL with varying numbers of trainable backbone parameters. The action decoder is implemented as a 16-layer DiT with 0.5B parameters. For further-trained VLM baselines, we include RoboBrain (Cao et al., 2025), VeBrain (Luo et al., 2025), and Cosmos-Reason1 (Azzolini et al., 2025), which are high-performing models further trained from Qwen2.5-VL on embodied reasoning and robotics datasets, as well as NORA (Hung et al., 2025), which is trained to predict FAST (Pertsch et al., 2025) tokenized actions (see Appendix B.4 for details).

**Results on different pre-trained VLM backbones.** Fig. 7 summarizes the effect of RS-CL when training VLA models from different pre-trained VLMs. RS-CL consistently improves the success rate across all backbones, demonstrating that our representation regularization generalizes beyond a particular backbone VLM. On SigLIP2, RS-CL yields larger improvements when the backbone is unfrozen, indicating that RS-CL can benefit from increased trainable capacity.

*Table 4.* **Ablation study on design choices.** Results report the average success rate (%) on RoboCasa-Kitchen with 300 demonstrations, analyzing the effect of different **(a)** distance definitions for soft-label supervision and **(b)** augmentation methods.

| Soft-label target | Avg. | Augmentation method | Avg. |
|---|---|---|---|
| Baseline (*i.e.*, no regularization) | 65.7 | No augmentation | 65.3 |
| No soft label (*i.e.*, InfoNCE) | 67.3 | Token cutoff | 66.3 |
| Next action sequence distance | 66.7 | Feature cutoff | 67.5 |
| Next single action distance | 66.8 | Span cutoff | 67.3 |
| Current state distance | **69.7** | View cutoff | **69.7** |

| (a) Soft-label target. | (b) Augmentation method. |
|---|---|

**Comparison to VLM training strategies.** Fig. 7 compares RS-CL with VLMs further trained on robotics datasets, for tasks such as visual grounding, embodied reasoning, and discretized action prediction. While such further-trained VLMs, when used as conditioning models, provide only limited and often inconsistent gains across backbone families, RS-CL consistently delivers larger improvements. It achieves higher success rates than any of these adapted models on both Qwen2.5-VL-3B and 7B, and further enhances their benefits when combined with them. Even for GR00T N1.5 VLM (NVIDIA GEAR, 2025), which is derived from Eagle 2.5 VLM (Chen et al., 2025) with enhanced grounding and reasoning capabilities, RS-CL provides additional gains. These results suggest that robotics-specific training alone may not fully close the gap between general-purpose VLM representations and the demands for action prediction, while RS-CL effectively bridges much of the gap.

## 3.3. Ablation Studies and Analyses

**Effect of soft-label supervision target.** In Table 4a, we observe that standard InfoNCE improves over the baseline without contrastive learning, demonstrating the effectiveness of our training framework, namely contrastive representation regularization for VLA models. However, alternative supervision signals (see Appendix B.3 for distance definitions) such as next action distances fall below vanilla InfoNCE. A plausible reason is that next actions have multimodal characteristics, and they themselves serves as the

*Table 5.* **KNN task-phase classification accuracy (%) on identical task trajectories across diverse scenes.** RS-CL enhances representation quality for task progress recognition under visual variations, narrowing the gap to ground-truth state performance.

| Features | Acc. (%) |
|---|---|
| Pre-trained VLM embeddings | 1.2 |
| Embeddings trained with action prediction | 20.3 |
| Embeddings trained with RS-CL | 22.9 |
| Ground-truth proprioceptive states | 25.6 |

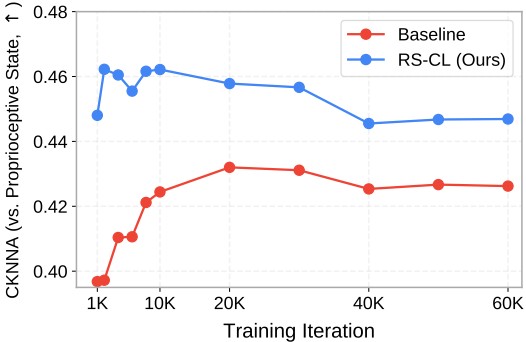

*Figure 8.* **Alignment to proprioceptive states.** We measure the alignment of conditioning representations inside trained VLA models, to the robot's proprioceptive states using CKNNA (Huh et al., 2024). RS-CL successfully improves the representation alignment to robot states of VLA models, compared to the model solely trained with action prediction loss.

prediction target, making it difficult to use as a reliable alignment signal. In contrast, the robot proprioceptive state provides a stable cue for representation alignment.

**Effect of representation augmentation strategy.** In Table 4b, we observe limited improvements from similar representation-level cutoff operations (Shen et al., 2020), while our proposed *view cutoff* achieves the highest success rate. This shows that simulating viewpoint variation is particularly beneficial for robust representation learning in multi-view robotic manipulation settings. This is in line with prior works, addressing the effects of utilizing multi-view data for representation learning (Weinzaepfel et al., 2022; Seo et al., 2023).

**Quantitative analysis of representation alignment.** We further analyze how RS-CL improves the alignment of VLM representations with robotic signals using KNN classification and CKNNA (Huh et al., 2024). As shown in Table 5, RS-CL improves task-phase classification, suggesting that the conditioning representation encodes state observations more reliably and is robust to visual background changes, thereby providing a more stable signal for the action decoder. Consistently, RS-CL increases representation similarity between learned embeddings and proprioceptive features (see Fig. 8), indicating that it reshapes the embedding space toward capturing control-relevant signals. Additional details for the experiments are provided in Appendix B.3.

## 4. Related Works

**Leveraging VLM representations for robot manipulation.** Vision-Language-Action (VLA) models have shown strong capabilities in robotic control by leveraging semantically enriched features from pre-trained Vision-Language Models (VLMs) (Zitkovich et al., 2023; Driess et al., 2023; Kim et al., 2024; Black et al., 2025b; Pertsch et al., 2025; Bjorck et al., 2025). A widely used architecture for VLA models consists of a pre-trained VLM and an action decoder with dedicated parameters (Black et al., 2025b; Bjorck et al., 2025; Shukor et al., 2025; Li et al., 2024; Zhou et al., 2025; Wen et al., 2025), training the VLM backbone with action prediction loss. Prior works have sought to further train VLMs for core knowledge of robot manipulation such as embodied reasoning and physical grounding (Cao et al., 2025; Luo et al., 2025; Azzolini et al., 2025; NVIDIA GEAR, 2025), or by discretized action prediction (Kim et al., 2025; Black et al., 2025a). Other methods jointly train the VLM with the action decoder on the aforementioned objectives. (Driess et al., 2025; Yang et al., 2025). Distinct from these approaches, our method does not rely on large-scale curated robotics datasets but instead improves VLM representations via a self-supervised objective.

**Contrastive representation learning.** Contrastive learning has been widely adopted to acquire transferable representations from high-dimensional inputs (Oord et al., 2018; Chen et al., 2020; He et al., 2020; Laskin et al., 2020; Radford et al., 2021). In robotics, contrastive objectives enable robust transfer of visuomotor policies, leveraging temporal consistency (Sermanet et al., 2018; Nair et al., 2022; Ma et al., 2023) or multi-view data (Seo et al., 2023). Recent works extend this idea to multimodal alignment (Rana et al., 2023; Myers et al., 2023; Lee et al., 2025), producing behaviorally grounded embeddings for control. Unlike prior approaches that apply contrastive learning as a separate representation learning stage, we integrate it directly into VLA training, enabling single-stage end-to-end optimization with the action prediction objective.

## 5. Conclusion

In this work, we present *Robot State-aware Contrastive Loss (RS-CL)*, a simple and effective regularization method that explicitly aligns representations with robot proprioceptive states. Our experiments demonstrate that RS-CL consistently improves VLA performances, particularly on tasks requiring reliable and precise positioning. These findings highlight the importance of shaping conditioning representations with physically grounded signals for accurate action prediction. We hope this work encourages further exploration of incorporating robot state and sensor signals, such as object pose or tactile feedback, to advance VLA models toward more precise and versatile robot control.

## Impact Statement

This paper presents a work to advance robot manipulation with vision-language-action models. The societal implications of this work are consistent with those commonly associated with research on robotic learning systems, including safe deployment and appropriate use in real-world environments. We do not foresee any unique ethical concerns arising from this work beyond those commonly encountered in research on robotic learning.

## Acknowledgments

This work was partly supported by Institute for Information & Communications Technology Planning & Evaluation (IITP) grant funded by the Korea government (MSIT) (RS-2019-II190075, Artificial Intelligence Graduate School Program (KAIST); RS-2024-00509279, Global AI Frontier Lab; RS-2025-02653113, High-Performance Research AI Computing Infrastructure Support at the 2 PFLOPS Scale), and RLWRLD Inc. We also thank Sihyun Yu for his valuable comments and suggestions in preparing an earlier version of the manuscript, and Suhyeok Jang for helpful support in conducting real-world experiments.

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

## A. Hardware Details and Computation Overhead

All experiments are conducted on a single node equipped with $2 \times$ NVIDIA A100-SXM4-80GB GPUs and 64 CPU cores. Unless otherwise noted, we use a global batch size of 64 and train for 60K optimization steps.

To quantify the additional cost introduced by RS-CL, we measure both floating point operations (FLOPs) and wall-clock training time for our fine-tuning experiment in RoboCasa-Kitchen. Using a FLOPs profiler, we measure the forward FLOPs per sample during training. Table 6 summarizes the compute characteristics for a single training run.

*Table 6.* Compute overhead of RS-CL. We report estimated forward FLOPs per sample and total training time for 60K steps with global batch size 64.

| Method | FLOPs / sample (forward) | Training time (hrs) |
| --- | --- | --- |
| Baseline | $2.576 \times 10^{12}$ | 23.06 |
| RS-CL | $2.909 \times 10^{12}$ | 23.49 |

The additional wall-clock cost introduced by RS-CL is negligible $(+1.25\%)$, because the *view-cutoff* augmentation operates directly on the VLM embeddings and RS-CL only adds a lightweight projection head and soft contrastive loss on top of the backbone forward pass. In particular, it does not require extra forward passes through the VLM backbone or longer token sequences, so the dominant compute costs of training remain essentially unchanged.

## B. Simulation Experiment Details

### B.1. Dataset

For RoboCasa-Kitchen, we use the publicly available dataset [1] containing 3000 demonstrations generated with Mimic-Gen (Mandlekar et al., 2023). For LIBERO, we use the publicly available dataset [2], consisting of all 270K samples from LIBERO-Spatial, LIBERO-Object, LIBERO-Goal, and LIBERO-Long, re-rendered by Kim et al. (2024).

### B.2. Training and Evaluation Details

For fine-tuning experiments on GR00T N1.5 (NVIDIA GEAR, 2025), we employ the publicly available pre-trained checkpoint [3]. We follow the original training and inference recipe of NVIDIA GEAR (2025), including the prior distribution $p(s) = \text{Beta}(\frac{a-s}{a}; 1.5, 1), a = 0.999$ for sampling the flow-matching timestep $s$ in equation 1. All models are trained with the *new_embodiment* tag. We omit the use of future tokens (Zheng et al., 2025), as they are beyond the scope of this work.

For RoboCasa-Kitchen, we train for 60K gradient steps with a global batch size of 64, using AdamW with a learning rate of 1e-4 under a cosine decay schedule and 3K warmup steps. For LIBERO, we adopt a smaller global batch size of 32, as this setting yields better performance in practice.

For $\pi_0$ and $\pi_0$-FAST, we use the pre-trained checkpoints [4] [5] to reproduce fine-tuned performance on RoboCasa-Kitchen. We train $\pi_0$ for 60K steps and $\pi_0$-FAST for 30K steps, both with a global batch size of 64. We set the learning rate to 2.5e-5 with cosine decay to 2.5e-6 and 1K warmup steps. At inference, we use an action horizon $H = 16$ and execute all actions without re-planning.

For RoboCasa-Kitchen, we evaluate all models with 1200 trials. For LIBERO, we evaluate 50 trials for each task, following Kim et al. (2024).

---

[1] https://huggingface.co/datasets/nvidia/PhysicalAI-Robotics-GR00T-X-Embodiment-Sim
[2] https://huggingface.co/datasets/physical-intelligence/libero
[3] https://huggingface.co/nvidia/GR00T-N1.5-3B
[4] gs://openpi-assets/checkpoints/pi0_base
[5] gs://openpi-assets/checkpoints/pi0_fast_base

## B.3. Analysis Details

**TCN implementation details.** TCN learns embeddings by pulling temporally adjacent observations together while pushing apart observations from distant timesteps. Since recent VLA models consume multi-view observations (NVIDIA GEAR, 2025; Black et al., 2025b) in a single forward pass, the multi-view TCN variant samples negatives from timesteps outside a temporal margin range, while positives are generated by zeroing out a randomly selected camera view. The single-view TCN variant follows the original formulation, drawing positives from a nearby temporal window and negatives from a distant temporal window. Following the original work (Sermanet et al., 2018), we set the temporal margin for defining positive/negative pairs to 0.2s.

**Soft label target distance definition.** For the ablation study on soft label targets in Sec. 3.3, we define distances as follows. For next single action and current state, we use Euclidean distance. For next action sequence, we use Dynamic Time Warping (DTW), which measures similarity between temporal sequences that may vary in speed. DTW requires an additional temperature hyperparameter $\gamma$, which we set to 10.0. The soft weight temperature $\beta$ and similarity temperature $\tau$ are fixed at 1.0 and 0.2, respectively.

**CKNNA measurement.** CKNNA (Huh et al., 2024) is a nearest-neighbor variant of kernel alignment (Kornblith et al., 2019). We randomly sample 10 trajectories per task in RoboCasa-Kitchen, totaling 240 trajectories. Each trajectory is processed with a window size of 16, yielding 4415 transitions. We extract the embeddings from the adapter module $f_\phi$ (used as conditioning inputs to the action decoder) along with the corresponding proprioceptive states. We follow the implementation of Huh et al. (2024) and report results with $k = 10$, measuring the alignment between proprioceptive states and conditional representations in the VLA model.

**KNN task-phase classification.** We perform a cross-scene KNN task-phase classification experiment on identical manipulation trajectories replayed in different visual environments. For each trajectory, we first align the executions across scenes using Dynamic Time Warping (DTW), and discretize the aligned time axis into 32 shared task-progress classes. Each timestep is assigned a scene-invariant phase label describing where it lies in the overall execution of the task. Given a trained model, we extract the conditioning representation at each timestep and evaluate how well it encodes task progress under visual changes by training a KNN classifier in this representation space to predict the phase labels. We compare four feature choices: (i) the pre-trained VLM representation without any robot-action training, (ii) the representation from a model trained only with the action prediction loss (baseline), (iii) the representation from a model trained with RS-CL, and (iv) the ground-truth proprioceptive state, which serves as an upper-bound reference for phase information. We use a KNN classifier with k = 5.

## B.4. Implementation Details for From-Scratch VLA Training

We attach a randomly initialized action decoder to various pre-trained VLMs, with a lightweight adapter module $f_\phi$ in between. Following NVIDIA GEAR (2025), we define $\text{VLM}(\mathbf{O}_t^V, \mathbf{c})$ as the hidden representation from layer 12 out of 36 layers for Qwen2.5-VL-3B variants and the GR00T N1.5 backbone. For Qwen2.5-VL-7B, we extract $\text{VLM}(\mathbf{O}_t^V, \mathbf{c})$ from layer 18 out of 28, which yields higher performance in our layer ablation study on LIBERO (see Table 7). For SigLIP, we instead use the final hidden representation as the condition embedding.

As the action decoder, we adopt a 16-layer DiT with 0.5B parameters. Empirically, we find that omitting a projection layer to reduce embedding dimensionality before conditioning improves performance (see Table 7). Accordingly, we do not apply such a layer. Instead, for Qwen2.5-VL-7B variants, we use a larger attention dimension that matches its hidden size $d_{\text{model}} = 3584$, while Qwen2.5-VL-3B uses $d_{\text{model}} = 2048$.

*Table 7.* **Hidden layer ablations on Qwen2.5-VL-7B backbone.** We report success rates (%) on the LIBERO benchmark, varying the hidden layer index used as the conditioning representation for VLA models trained from scratch.

| Layer | Spatial | Object | Goal | Long | Avg. |
|---|---|---|---|---|---|
| 12 (with projection) | 87.4 | 94.2 | 41.8 | 40.4 | 66.0 |
| 18 (with projection) | 86.8 | 83.4 | 61.6 | 44.0 | 69.0 |
| **18 (no projection)** | 85.2 | 89.4 | 73.2 | 36.2 | **71.0** |
| 24 (with projection) | 85.2 | 89.4 | 73.2 | 36.2 | 57.0 |

# C. Real-Robot Experiment Details

## C.1. Hardware Platform

We use Franka Research 3, a 7-DoF robotic arm equipped with a Robotiq 2F-85 gripper, following the setup of Khazatsky et al. (2024). For visual perception, we utilize the dual camera setup: a movable Stereolabs ZED 2 provides a global view, and a wrist-mounted ZED Mini captures a close-range view. Teleoperated demonstrations are collected using an Oculus Quest 2, and we log time-synchronized RGB images, joint states, and gripper width for training and evaluation. Demonstrations are recorded at 10 Hz.

## C.2. Real-Robot Tasks

The in-domain and generalization tasks (visual, physical generalization, and language grounding) along with their corresponding prompts and representative key frames from the real-world evaluation, are shown in Fig. 9– 13.

**In-domain tasks.** We introduce four pick-and-place tasks (Box to Bowl, Box to Plate, Basket to Bowl, Plate to Basket), with varied objects (teddy bear, blue cube, blue cup, yellow sponge) for each task (see Fig. 9).

**Visual generalization.** We use in-domain objects differing in color (*e.g.*, changing a blue cube to a green cube, or a yellow sponge to a blue sponge), and introduce background variations by changing the tabletop covering or the target container (see Fig. 10).

**Physical generalization.** We evaluate with unseen objects not used in training, including a yellow banana, purple grapes, red strawberry, and a yellow cup. The yellow cup has a different shape and texture from the blue cup used in training. (see Fig. 11).

**Language grounding.** We place two objects that are seen at training at the pick-up location, and specify which one to pick and place at target location (see Fig. 12).

**Lighting generalization.** We evaluate on in-domain tasks under significantly darker lighting conditions than those used in training (see Fig. 13).

## C.3. Real-World Training and Evaluation Details

**Dataset.** We collect 60 demonstrations for each pick-and-place task and close lid task, resulting in 300 expert demonstrations.

**Training.** We jointly train a model with the 4 pick-and-place tasks, and another model for the close-lid task. For pick-and-place, we employ a cartesian action space with proprioceptive states, and for the close-lid task we use a joint action space to cover various configurations in manipulation.

**Evaluation.** For real-robot evaluation, we report the average success rate over 24 trials for each pick-and-place task, with varied objects. In the close-lid task, outcomes are classified as full success (lid fully closed), partial success (partially closed), or failure (not closed). For physical generalization, we evaluate with unseen objects (yellow banana, purple grapes, red strawberry, yellow cup), with success defined as the accurate completion of the pick-and-place. We define language following as whether the gripper approaches the correct object, and task success as completing the instructed pick-and-place.

**Box to Bowl :**
"Pick up the **[**teddy bear / blue cube / blue cup / yellow sponge**]** on the brown box and place it in the golden bowl."

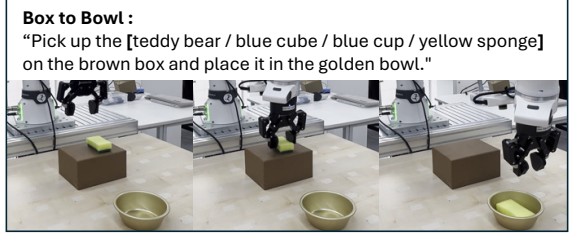

**Box to Plate :**
"Pick up the **[**teddy bear / blue cube / blue cup / yellow sponge**]** on the brown box and place it in the white plate."

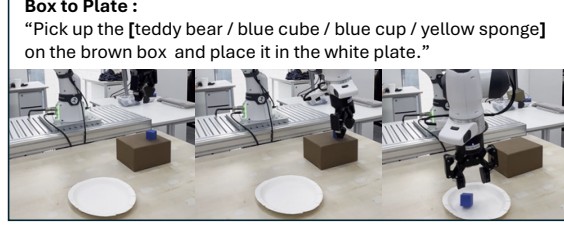

**Basket to Bowl :**
"Pick up the **[**teddy bear / blue cube / blue cup / yellow sponge**]** in the white basket and place it in the black bowl."

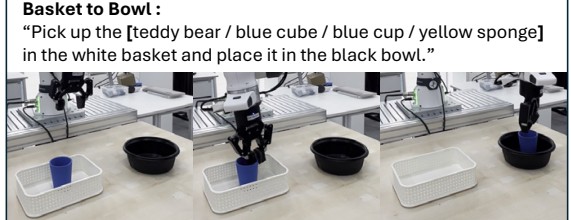

**Plate to Basket :**
"Pick up the **[**teddy bear / blue cube / blue cup / yellow sponge**]** in the white plate and place it in the white basket."

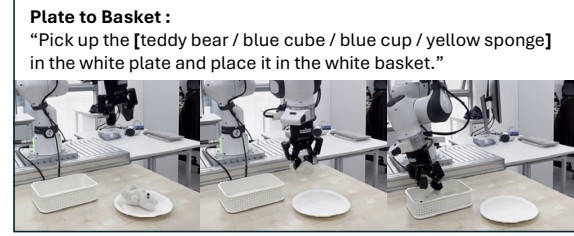

**Close Lid :**
"Pick up the lid and close it on the pot."

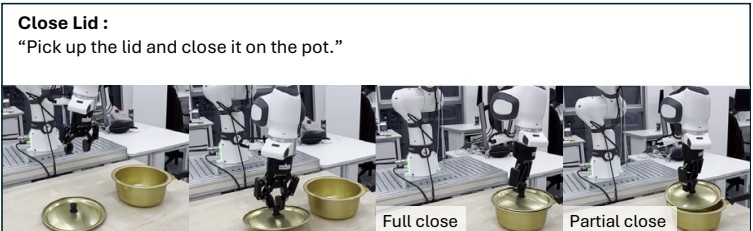

*Figure 9.* Real-world in-domain tasks.

**Box to Bowl :**
"Pick up the **green cube** on the brown box and place it in the golden bowl."

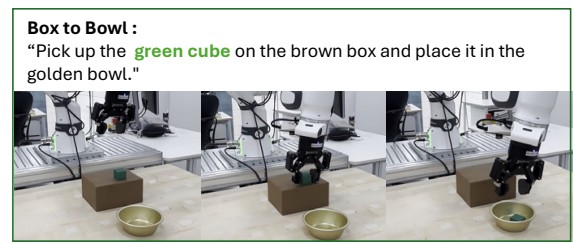

**Basket to Bowl :**
"Pick up the **blue sponge** in the white basket and place it in the black bowl."

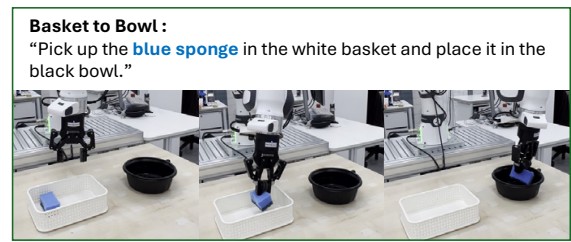

**Box to Plate : (+ Tabletop Background changed)**
"Pick up the **[**teddy bear / blue cube / blue cup / yellow sponge**]** on the brown box and place it in the **black plate**."

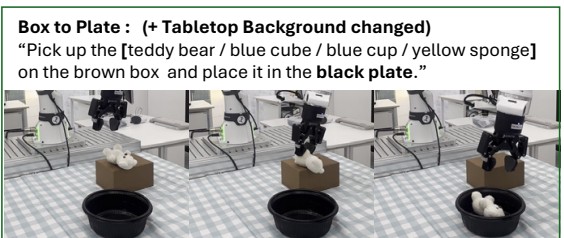

**Plate to Basket : (+ Tabletop Background changed)**
"Pick up the **[**teddy bear / blue cube / blue cup / yellow sponge**]** in the white plate and place it in the **brown basket**."

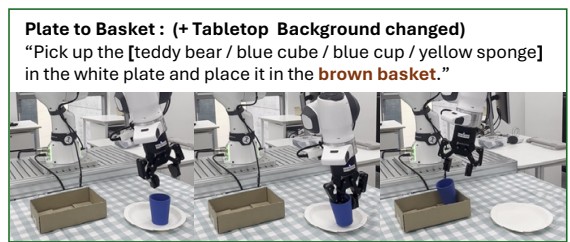

*Figure 10.* Real-world visual generalization tasks.

**Box to Bowl :**
"Pick up the **yellow banana** on the brown box and place it in the gold bowl."

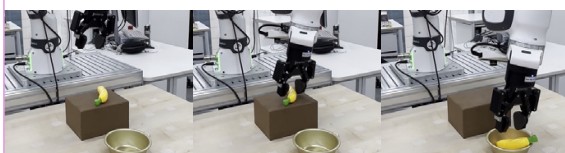

**Box to Bowl :**
"Pick up **purple grapes** on the brown box and place it in the gold bowl."

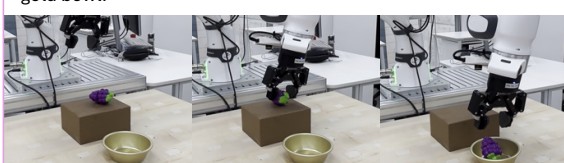

**Box to Bowl :**
"Pick up the **red strawberry** on the brown box and place it in the gold bowl."

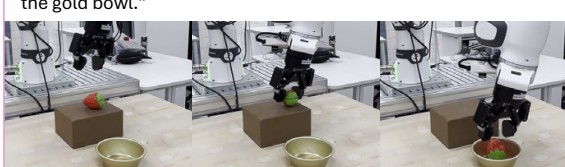

**Basket to Bowl :**
"Pick up the **yellow cup** in the white basket and place it in the black bowl."

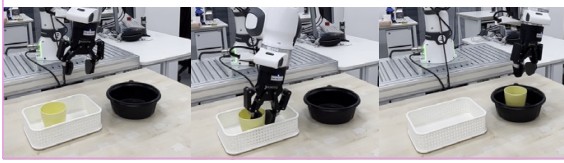

*Figure 11.* Real-world physical generalization tasks.

**Box to Bowl :**
"Pick up the **[teddy bear** / blue cup**]** on the brown box and place it in the gold bowl."

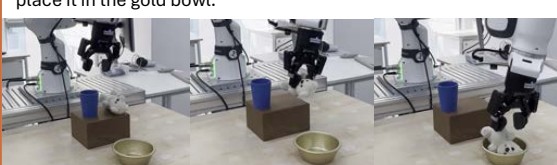

**Box to Bowl :**
"Pick up the **[**teddy bear / **blue cup]** on the brown box and place it in the gold bowl."

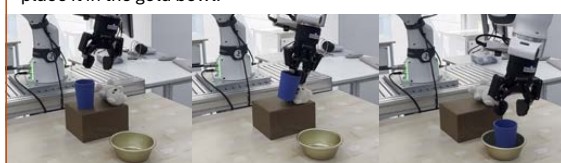

**Basket to Bowl :**
"Pick up the **[teddy bear /** yellow sponge**]** in the white basket and place it in the black bowl."

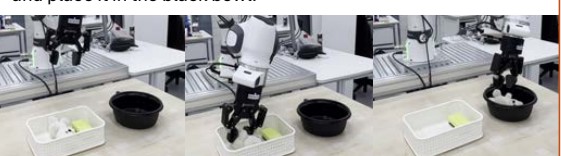

**Box to Plate :**
"Pick up the **[**teddy bear / **blue cup]** on the brown box and place it in the white plate."

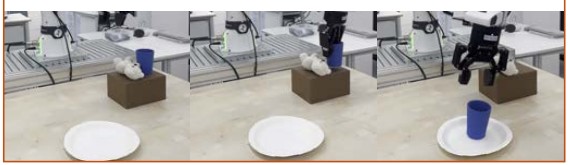

**Plate to Basket :**
"Pick up the **[**teddy bear **/ yellow sponge]** in the white plate and place it in the white basket."

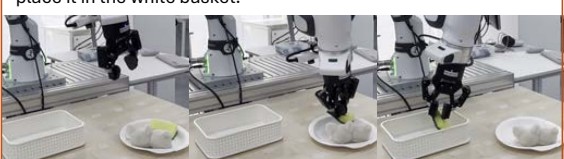

**Plate to Basket :**
"Pick up the **[teddy bear** / yellow sponge**]** in the white plate and place it in the white basket."

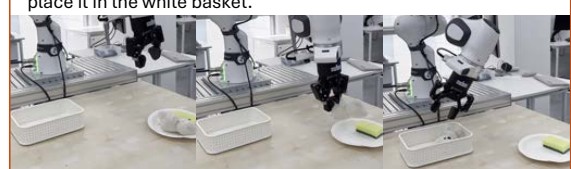

*Figure 12.* Real-world language grounding tasks.

**Box to Plate :**
"Pick up the **[**teddy bear / blue cube / blue cup / yellow sponge**]** on the brown box and place it in the white plate."

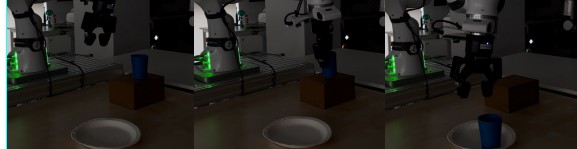

**Plate to Basket :**
"Pick up the **[**teddy bear / blue cube / blue cup / yellow sponge**]** in the white plate and place it in the white basket."

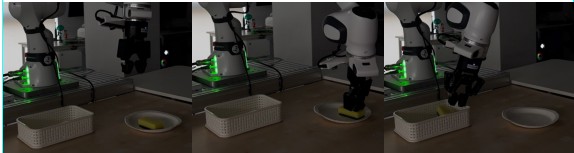

*Figure 13.* Real-world lighting variation tasks.

# D. Hyperparameters

We detail the hyperparameter choices used for RS-CL and analyze the sensitivity of the method to its main components. Unless otherwise specified, we initialize the weighting coefficient $\lambda$ for $\mathcal{L}_{\text{RS-CL}}$ to 1.0 and decay it to 0 following a cosine schedule over the total training steps. This design emphasizes representation refinement in the early stages of training, while gradually shifting the focus toward accurate action prediction in later stages.

We set the similarity temperature $\tau$ to 0.2 and the distance-based soft weighting temperature $\beta$ to 1.0 across all experiments. As summarized in Table 8, RS-CL exhibits stable performance over a wide range of hyperparameter values, including different $\lambda$ schedules, temperature settings, projection dimensions, batch sizes, and random seeds. These results indicate that the proposed method is robust and does not rely on precise hyperparameter tuning to achieve strong performance.

*Table 8.* **Hyperparameter ablations.** Average success rate (%) on RoboCasa-Kitchen with 30 demonstrations. Baseline performance without RS-CL is 48.2. The selected settings in the main experiments are highlighted in **bold**.

| Hyperparameter | Setting | Avg. SR |
|---|---|---|
| Baseline | - | 48.2 |
| $\lambda$ schedule | $1.0 \rightarrow 0$ (cosine) 
 Fixed ($\lambda = 1.0$ / 0.5) | **53.0** 
 50.7 / 51.0 |
| Similarity temp. $\tau$ | 0.01 / 0.02 / 0.05 / 0.1 / 1.0 | 51.6 / 53.0 / 52.0 / **53.3** / 51.1 |
| Distance temp. $\beta$ | 0.1 / 1.0 / 10.0 | 51.2 / **53.0** / 49.8 |
| Projection dim | 64 / 128 / 256 | 50.9 / **53.0** / 51.2 |
| Batch size | Baseline (32 / 64) 
 RS-CL (32 / 64) | 48.4 / 48.2 
 51.5 / **53.0** |
| Training Seeds | Baseline (0 / 7 / 42) 
 RS-CL (0 / 7 / 42) | 49.2 / 48.8 / 48.2 
 54.7 / 51.3 / **53.0** |

# E. Further Analysis

## E.1. Detailed Visualizations of VLM Representations

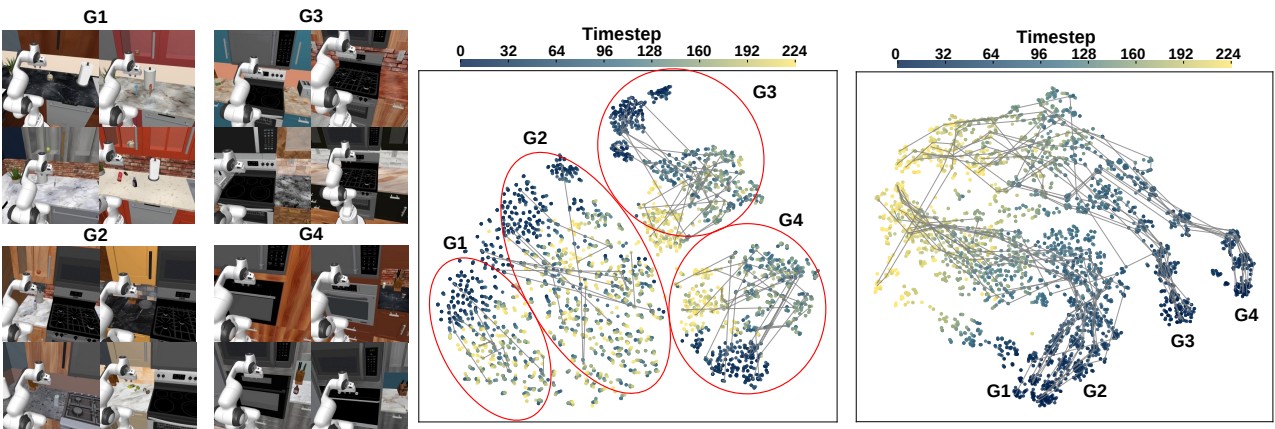

*(a)* Pre-trained VLM representations.

*(b)* RS-CL aligned representations.

*Figure 14.* **Detailed visualizations of VLM representations. (a)** We visualize VLM embeddings of robot episodes performing the same task "Open the microwave / cabinet door" across different scenes in RoboCasa-Kitchen. Pre-trained VLM representations form distinct clusters primarily based on visual appearance, and the task/timestep progress trajectories (*i.e.*, gray lines starting from blue to yellow dots) are not consistently aligned across these clusters. **(b)** With RS-CL, the representations still preserve scene-dependent grouping, but the task progress becomes geometrically aligned across groups (*i.e.*, from bottom toward the top-left region), indicating that episodes from different environments share a common progression direction in the embedding space.

We provide a more detailed explanation of Fig. 2 using the additional visualizations in Fig. 14. In the pre-trained VLM representation (Fig. 14a, we observe four prominent clusters (*i.e.*, G1–G4) that are clearly grouped by scene layout. Specifically, G1 corresponds to scenes where a flat, wide tabletop occupies most of the space in front of the robot; G2 to scenes with a tabletop in front and a stove or burner at the front right; G3 to scenes dominated by a stove or burner directly in front of the robot; and G4 to scenes where an oven is positioned in front of the robot. These clusters are thus primarily induced by the most visually dominant objects in the scene. However, the underlying task, "Open the microwave / cabinet door," mainly requires the robot arm to move upward and reach an overhead door in front of the robot, which is largely independent of these dominant background objects. Consistent with this mismatch, the timestep-indexed task progress trajectories (*i.e.*, gray lines from blue to yellow) do not align across clusters, indicating that the pre-trained VLM embeddings are organized by visual appearance rather than by control-relevant task progress.

In contrast, Fig. 14b shows the condition representations of the VLA model trained with RS-CL. While episode embeddings from different scenes still form partially separated groups, the common task progress becomes geometrically aligned across these groups. Trajectories consistently evolve from the bottom toward the top-left region of the embedding space. This suggests that RS-CL reshapes the conditioning representation to be more robot-state centric, so that episodes at similar task phases are aligned even when they come from visually distinct environments, while still preserving the semantic grouping from the pre-trained VLM.

## E.2. Quantitative Analysis of Representation Structure

To complement the qualitative visualizations in Fig. 14, we quantify how the representations are organized by measuring KNN classification accuracy ($k = 5$) on the embeddings. We report (i) *scene classification accuracy*, predicting the scene cluster (G1–G4 in Fig. 14a), and (ii) *task-progress classification accuracy*, predicting the discretized task phase. We additionally report results on the raw proprioceptive state as a natural reference for task-progress structure.

As shown in Table 9, pre-trained VLM embeddings achieve near-perfect scene accuracy (99.6%) but near-chance task-progress accuracy (1.2%), confirming that they are organized by visual appearance rather than control-relevant task progress. RS-CL substantially improves task-progress structure (1.2% → 22.9%), approaching the proprioceptive-state reference (25.6%), while largely preserving scene information (93.6%). This shows that RS-CL effectively transfers the task-progress structure implicit in proprioceptive states into the VLM representation, consistent with the trajectory alignment in Fig. 14b.

*Table 9.* **KNN classification accuracy (%) on the representation space.** Pre-trained VLM representations are dominated by scene-dependent visual appearance and carry almost no task-progress information. RS-CL substantially improves task-progress structure, nearly matching the proprioceptive-state reference, while preserving most of the scene information from the pre-trained VLM.

| Representation | Scene Acc.(%) | Task progress Acc.(%) |
|---|---|---|
| Pre-trained VLM embeddings | 99.6 | 1.2 |
| Embeddings trained with RS-CL | 93.6 | 22.9 |
| Ground-truth proprioceptive states | 53.1 | 25.6 |

### E.3. Adaptation to another VLA Architecture

To verify that RS-CL remains effective regardless of the underlying action modeling paradigm of the baseline VLA model, we apply it to $\pi_0$-FAST (Pertsch et al., 2025), an autoregressive VLA model that predicts action tokens via next-token prediction instead of flow matching. Under the RoboCasa-Kitchen fine-tuning setting, $\pi_0$-FAST with RS-CL improves performance across all demonstration counts (see Table 10). These results suggest that, as long as the VLA model conditions its policy on a large-scale pre-trained VLM backbone, RS-CL is a broadly applicable and beneficial regularization strategy, independent of the specific action modeling design.

*Table 10.* **RoboCasa-Kitchen benchmark success rate (%).** Results include fine-tuned performance of $\pi_0$-FAST (Pertsch et al., 2025), an autoregressive VLA model and GR00T N1.5 (NVIDIA GEAR, 2025), an flow-matching VLA model, with RS-CL. Best results within the same backbone indicated in **bold**.

| Method | 30 demos | 100 demos | 300 demos |
|---|---|---|---|
| $\pi_0$-FAST (Autoregressive) | 29.8 | 60.2 | 63.6 |
| **+ RS-CL (Ours)** | **33.2** | **61.1** | **65.2** |
| GR00T N1.5 (Flow-Matching) | 48.2 | 63.9 | 65.7 |
| **+ RS-CL (Ours)** | **53.0** | **67.2** | **69.7** |

### E.4. Applicability across Proprioceptive State Configurations

Different robots naturally provide different proprioceptive state representations. To examine the applicability of RS-CL across such variations, we evaluate it under different state configurations on RoboCasa-Kitchen (Nasiriany et al., 2024) and on more complex embodiments in DexMimicGen (Jiang et al., 2025), while keeping the rest of the framework unchanged.

**State definition on RoboCasa-Kitchen.** We compare two natural choices for the proprioceptive state: end-effector pose (the default in the baseline VLA framework) and joint position. As shown in Table 11(a), RS-CL consistently improves over the baseline under both choices, with joint position yielding slightly stronger gains. This indicates that the benefit does not depend on a particular state parameterization.

**Higher-dimensional state spaces on DexMimicGen.** We further evaluate RS-CL on more complex embodiments in DexMimicGen: bimanual Panda arms with two Inspire Hands (38-dim state) and the GR-1 humanoid platform performing upper-body manipulation (36-dim state). As shown in Table 11(b), RS-CL provides consistent improvements in both settings, showing that its benefit extends to higher-dimensional and more complex proprioceptive configurations.

*Table 11.* **RS-CL across diverse proprioceptive state configurations.** RS-CL consistently improves over the baseline across (a) different state definitions on RoboCasa-Kitchen and (b) more complex embodiments with higher-dimensional state spaces on DexMimicGen.

*(a)* Success rate (%) on RoboCasa-Kitchen (30 demos).

| Method | Success rate |
|---|---|
| Baseline | 48.2 |
| **+ RS-CL (EEF pose)** | 53.0 |
| **+ RS-CL (Joint position)** | **53.4** |

*(b)* Success rate (%) on DexMimicGen.

| Method | Bimanual Panda + Hands | GR-1 Humanoid |
|---|---|---|
| Baseline | 52.7 | 68.9 |
| **+ RS-CL (Ours)** | **57.3** | **73.1** |

## E.5. Task-wise Analysis

To better understand where RS-CL contributes most, we break down the RoboCasa-Kitchen success rates by task group: pick-and-place, open-and-close, and others (*e.g.*, twisting knobs or insertion). As shown in Table 12, RS-CL provides the largest gains on pick-and-place tasks ($+4.2$ to $+11.4\%$) and on twisting/insertion tasks ($+3.8$ to $+5.4\%$), with relatively smaller improvements on open-and-close tasks ($-1.3$ to $+1.0\%$).

*Table 12.* **RoboCasa-Kitchen success rate (%) by task group.** Main numbers are results for 30 demonstrations, with 300-demonstration results in parentheses. Full per-task results are provided in Table 13.

| Method | Pick and Place | Open and Close | Others (*e.g.*, twisting, insertion) |
| --- | --- | --- | --- |
| Baseline | 30.1 (55.6) | **71.3** (84.7) | 48.2 (62.6) |
| **+ RS-CL (Ours)** | **41.5 (59.8)** | 70.0 (**85.7**) | **52.0 (68.0)** |

We attribute this pattern to structural differences across task groups. Open-and-close tasks involve a constrained action space (*e.g.*, swinging a door or drawer along a fixed axis), where the baseline representation already performs strongly and leaves limited room for improvement. In contrast, pick-and-place tasks involve diverse state-to-action mappings due to varying initial and goal placements, and twisting/insertion tasks require precise state-dependent control near contact points. RS-CL's state-aware representation provides more informative supervision in these cases, translating into larger performance gains.

## E.6. Discussion

**Limitations.** While RS-CL explicitly leverages proprioceptive states to align the representation space, it does not incorporate further signals in robotic manipulation, such as object poses or contact forces. These modalities often provide complementary information that is captured by robot's proprioception state. Extending RS-CL to integrate such modalities into the representations, represents a promising direction for future research.

**Future directions.** One promising extension is to apply RS-CL to settings with more complex proprioceptive spaces, such as humanoid robots or dexterous hand manipulation tasks. These domains involve high-dimensional and complex state representations, where aligning VLM embeddings with proprioceptive signals may be even more beneficial for accurate action prediction.

# F. More Quantitative Results

In this section, we report detailed task-wise results of our main experiments. Table 13 reports RoboCasa-Kitchen success rates of GR00T N1.5 trained with and without RS-CL, Table 14 reports our reproduced results of $\pi_0$ and $\pi_0$-FAST on the same benchmark, and Table 15 reports task-wise results of the from-scratch experiments across various VLM backbones.

*Table 13.* **Detailed results on RoboCasa-Kitchen.** Task-wise success rates of GR00T N1.5 (NVIDIA GEAR, 2025) trained with, and without RS-CL, by different number of demonstrations.

| Task | GR00T N1.5 ($\mathcal{L}_{FM}$) | | | GR00T N1.5 ($\mathcal{L}_{FM} + \lambda\mathcal{L}_{RS-CL}$) | | |
|---|---|---|---|---|---|---|
| | 30 demos | 100 demos | 300 demos | 30 demos | 100 demos | 300 demos |
| RoboCasa Kitchen (24 tasks, PnP = Pick-and-Place) | | | | | | |
| Close Double Door | 44.0 | 86.0 | 80.0 | 54.0 | 78.0 | 86.0 |
| Close Drawer | 96.0 | 96.0 | 96.0 | 96.0 | 96.0 | 96.0 |
| Close Single Door | 98.0 | 94.0 | 98.0 | 88.0 | 98.0 | 98.0 |
| Coffee Press Button | 70.0 | 82.0 | 90.0 | 86.0 | 94.0 | 92.0 |
| Coffee Serve Mug | 64.0 | 72.0 | 58.0 | 74.0 | 66.0 | 70.0 |
| Coffee Setup Mug | 28.0 | 34.0 | 24.0 | 30.0 | 54.0 | 46.0 |
| Open Double Door | 80.0 | 92.0 | 82.0 | 72.0 | 80.0 | 84.0 |
| Open Drawer | 46.0 | 58.0 | 74.0 | 44.0 | 54.0 | 76.0 |
| Open Single Door | 64.0 | 58.0 | 78.0 | 66.0 | 60.0 | 74.0 |
| PnP from Cab $\rightarrow$ Counter | 28.0 | 42.0 | 54.0 | 38.0 | 54.0 | 60.0 |
| PnP from Counter $\rightarrow$ Cab | 36.0 | 54.0 | 54.0 | 40.0 | 58.0 | 68.0 |
| PnP from Counter $\rightarrow$ Microwave | 30.0 | 36.0 | 32.0 | 34.0 | 40.0 | 40.0 |
| PnP from Counter $\rightarrow$ Sink | 28.0 | 66.0 | 58.0 | 40.0 | 60.0 | 68.0 |
| PnP from Counter $\rightarrow$ Stove | 38.0 | 60.0 | 66.0 | 38.0 | 74.0 | 72.0 |
| PnP from Microwave $\rightarrow$ Counter | 24.0 | 44.0 | 50.0 | 46.0 | 50.0 | 48.0 |
| PnP from Sink $\rightarrow$ Counter | 40.0 | 52.0 | 60.0 | 54.0 | 62.0 | 68.0 |
| PnP from Stove $\rightarrow$ Counter | 22.0 | 60.0 | 68.0 | 42.0 | 66.0 | 54.0 |
| Turn Off Microwave | 62.0 | 86.0 | 94.0 | 62.0 | 84.0 | 94.0 |
| Turn Off Sink Faucet | 72.0 | 86.0 | 92.0 | 70.0 | 94.0 | 88.0 |
| Turn Off Stove | 10.0 | 14.0 | 28.0 | 10.0 | 8.0 | 28.0 |
| Turn On Microwave | 44.0 | 58.0 | 44.0 | 48.0 | 72.0 | 66.0 |
| Turn On Sink Faucet | 60.0 | 90.0 | 86.0 | 72.0 | 90.0 | 90.0 |
| Turn On Stove | 34.0 | 56.0 | 32.0 | 36.0 | 58.0 | 36.0 |
| Turn Sink Spout | 38.0 | 58.0 | 78.0 | 32.0 | 62.0 | 70.0 |
| **Average** | **48.2** | **63.9** | **65.7** | **53.0** | **67.2** | **69.7** |

*Table 14.* **Detailed results on RoboCasa-Kitchen.** Task-wise success rates (%) of reproduced $\pi_0$ (Black et al., 2025b) and $\pi_0$-FAST (Pertsch et al., 2025), by different number of demonstrations.

| Task | $\pi_0$ | | | $\pi_0$-FAST | | |
|---|---|---|---|---|---|---|
| | 30 demos | 100 demos | 300 demos | 30 demos | 100 demos | 300 demos |
| RoboCasa Kitchen (24 tasks, PnP = Pick-and-Place) | | | | | | |
| Close Double Door | 68.0 | 86.0 | 86.0 | 44.0 | 84.0 | 78.0 |
| Close Drawer | 94.0 | 94.0 | 96.0 | 84.0 | 96.0 | 94.0 |
| Close Single Door | 94.0 | 98.0 | 96.0 | 84.0 | 90.0 | 72.0 |
| Coffee Press Button | 66.0 | 80.0 | 88.0 | 20.0 | 82.0 | 90.0 |
| Coffee Serve Mug | 80.0 | 66.0 | 64.0 | 44.0 | 66.0 | 68.0 |
| Coffee Setup Mug | 20.0 | 32.0 | 38.0 | 2.0 | 34.0 | 38.0 |
| Open Double Door | 92.0 | 90.0 | 84.0 | 26.0 | 68.0 | 78.0 |
| Open Drawer | 44.0 | 56.0 | 62.0 | 36.0 | 58.0 | 68.0 |
| Open Single Door | 58.0 | 64.0 | 70.0 | 44.0 | 70.0 | 66.0 |
| PnP Cab $\rightarrow$ Counter | 14.0 | 22.0 | 18.0 | 12.0 | 22.0 | 30.0 |
| PnP Counter $\rightarrow$ Cab | 32.0 | 44.0 | 46.0 | 8.0 | 58.0 | 48.0 |
| PnP Counter $\rightarrow$ Microwave | 26.0 | 30.0 | 18.0 | 10.0 | 32.0 | 20.0 |
| PnP Counter $\rightarrow$ Sink | 32.0 | 44.0 | 58.0 | 2.0 | 46.0 | 56.0 |
| PnP Counter $\rightarrow$ Stove | 14.0 | 32.0 | 60.0 | 10.0 | 50.0 | 64.0 |
| PnP Microwave $\rightarrow$ Counter | 16.0 | 20.0 | 24.0 | 4.0 | 38.0 | 46.0 |
| PnP Sink $\rightarrow$ Counter | 22.0 | 24.0 | 66.0 | 12.0 | 56.0 | 62.0 |
| PnP Stove $\rightarrow$ Counter | 10.0 | 46.0 | 44.0 | 18.0 | 62.0 | 60.0 |
| Turn Off Microwave | 64.0 | 84.0 | 96.0 | 68.0 | 98.0 | 96.0 |
| Turn Off Sink Faucet | 72.0 | 86.0 | 94.0 | 48.0 | 76.0 | 94.0 |
| Turn Off Stove | 14.0 | 10.0 | 22.0 | 0.0 | 18.0 | 22.0 |
| Turn On Microwave | 58.0 | 82.0 | 70.0 | 52.0 | 68.0 | 88.0 |
| Turn On Sink Faucet | 80.0 | 82.0 | 86.0 | 40.0 | 66.0 | 74.0 |
| Turn On Stove | 26.0 | 68.0 | 42.0 | 12.0 | 52.0 | 38.0 |
| Turn Sink Spout | 50.0 | 68.0 | 72.0 | 36.0 | 54.0 | 76.0 |
| **Average** | **47.8** | **58.7** | **62.5** | **29.8** | **60.2** | **63.6** |

*Table 15.* **Detailed results of from-scratch experiments.** Task success rate (%) on the RoboCasa-Kitchen benchmark trained with 300 demonstrations. All models train a VLA from scratch, starting from each pre-trained VLM backbone. Best results within the same backbone indicated in **bold**.

| Backbone Model | Success Rate | | |
|---|---|---|---|
| | PnP | Others | Avg. |
| Qwen2.5-VL-3B (Bai et al., 2025) | 2.5 | 8.6 | 6.6 |
| **+ RS-CL (Ours)** | **3.5** | **16.0** | **11.8** |
| NORA (Hung et al., 2025) | 1.5 | 11.4 | 8.1 |
| **+ RS-CL (Ours)** | **3.5** | **23.3** | **16.7** |
| RoboBrain2.0-3B (Cao et al., 2025) | 2.8 | 13.9 | 10.2 |
| **+ RS-CL (Ours)** | **5.8** | **19.6** | **15.0** |
| Qwen2.5-VL-7B (Bai et al., 2025) | 2.5 | 12.4 | 9.1 |
| **+ RS-CL (Ours)** | **9.8** | **21.1** | **17.3** |
| RoboBrain2.0-7B (Cao et al., 2025) | 2.3 | 12.8 | 9.3 |
| **+ RS-CL (Ours)** | **12.0** | **25.9** | **21.3** |
| VeBrain-7B (Luo et al., 2025) | 3.0 | 10.9 | 8.3 |
| **+ RS-CL (Ours)** | **7.8** | **20.3** | **17.6** |
| Cosmos-Reason-7B (Azzolini et al., 2025) | 1.0 | 5.5 | 4.0 |
| **+ RS-CL (Ours)** | **7.3** | **15.9** | **13.0** |
| SigLIP2 (Tschannen et al., 2025) | 0.3 | 2.9 | 2.0 |
| **+ RS-CL (Ours)** | **0.8** | **3.5** | **2.6** |
| SigLIP2, unfrozen backbone | 3.3 | 4.4 | 4.0 |
| **+ RS-CL (Ours)** | **17.3** | **12.5** | **14.1** |
| GR00T N1.5 VLM (NVIDIA GEAR, 2025) | 37.5 | 62.0 | 53.8 |
| **+ RS-CL (Ours)** | **37.8** | **66.3** | **56.8** |

