# OpenReview forum: "Contrastive Representation Regularization for Vision-Language-Action Models"
_ICML.cc/2026/Conference — ICML 2026 regular_

### Official Review · Reviewer_LdE5 · 2026-03-10

**Soundness:** 3
**Presentation:** 3
**Significance:** 3
**Originality:** 2
**Overall Recommendation:** 4
**Confidence:** 4

**Summary:**

This paper proposes Robot State-aware Contrastive Loss (RS-CL), a lightweight regularizer for Vision Language Action (VLA) models. The key idea is to reshape VLM representations using continuous proprioceptive signals as soft supervision in a weighted contrastive loss. A representation level augmentation enables efficient contrastive pairing without additional vision forward passes. Experiments on RoboCasa, LIBERO, and real-robot manipulation tasks show consistent improvements over standard VLAs.

**Compliance With Llm Reviewing Policy:**

Affirmed.

**Final Justification:**

Authors addressed stability, diverse proprioceptive signals, and complex embodiment concerns (GR-1, 38-dim bimanual); robustness demonstrated across settings; I maintain 4 as the contribution is solid despite moderate novelty.

**Key Questions For Authors:**

1. Can RS-CL work with any proprioceptive signal? For example, joint angles, velocities? Are there constraints on scale, smoothness, or dimensionality for the distance-based weighting to remain meaningful?
2. Can RS-CL hurt performance? For instance, tasks where visual cues are more informative than state proximity.
3. Does RS-CL affect the semantic structure of the VLM embedding space? For example, does it change language conditioned generalization or instruction following behavior independent of control performance?

**Limitations:**

Yes

**Strengths And Weaknesses:**

**Strengths**
1. The idea is intuitive if VLM features underweight robot state, explicitly shaping the representation using proprioception makes sense.
2. The view-cutoff trick is practical and avoids extra forward passes, so the added compute cost is minimal.
3. The paper includes reasonable ablations on contrastive targets, augmentations, and hyperparameters.

**Weakness**
1. The main tables report single numbers without standard deviations or multiple seeds, so it is hard to judge how stable the improvements are.
2. The proprioceptive signal is fairly narrow (mostly end-effector pose). It is unclear how the method behaves with noisier or different state inputs.
3. The impact on language understanding is not clearly isolated, improvements in task success are shown, but it is not fully clear whether semantic capabilities are preserved or altered.

---

> ### Author Rebuttal · Authors · 2026-03-31
>
> Dear Reviewer LdE5,
>
> We sincerely appreciate your efforts and comments to improve the manuscript. We respond to your comment in what follows.
>
> ---
> **[W1] Stability of experiment results**
>
> To address this concern, we will include standard errors in the main tables in the revised manuscript.  The baseline achieves 48.2% (1.44) / 63.9% (1.38) / 65.7% (1.37), while RS-CL achieves 53.0% (1.44) / 67.2% (1.36) / 69.7% (1.33) across the three demo settings in RoboCasa-Kitchen. For the real-world experiments, the baseline and RS-CL achieve 45.0% (4.54) and 58.3% (4.50), respectively, as shown in Fig. 6.
> We also ran three training seeds for the 30-demo setting (Table 8). The baseline achieves 49.2% / 48.8% / 48.2% for seeds 0 / 7 / 42, while RS-CL achieves 54.7% / 51.3% / 53.0%. RS-CL is better than the baseline in all three runs, supporting that the improvement is stable across training seeds.
>
> ---
> **[W2, Q1] Applicability to different proprioceptive signals**
>
> RS-CL is not tied to a specific proprioceptive signal. To support this, we replaced the state used for the distance-based weighting with different signals, including end-effector pose (3D position, 6D rotation, 2D gripper pose, as in the main paper), joint positions (7D joint positions, 2D gripper pose), and their combination (18D), while keeping the rest of the framework unchanged. Across all choices, RS-CL consistently improves over the baseline.
> \begin{array}{lc}
> \hline
> \text{Method} & \text{RoboCasa-Kitchen (\\%)} \newline
> \hline
> \text{Baseline} & 48.2 \newline
> \text{RS-CL (EEF pose)} & 53.0 \newline
> \text{RS-CL (Joint position)} & 53.4 \newline
> \text{RS-CL (EEF + Joint position)} & 51.5 \newline
> \hline
> \end{array}
> The key requirement is that the resulting distance remain meaningful for control-relevant similarity. In practice, we ensure this by applying min-max normalization to each state dimension into [-1, 1] before computing the weights.
>
> We also refer the reviewer to our responses to [W3] of reviewer UWEC and [Q3] of reviewer QCsh, where we provide additional results on richer state spaces, including bimanual arm with hands and the GR-1 humanoid setting.
>
> ---
> **[W3, Q3] Regarding semantic capabilities and instruction-following behavior**
>
> To directly evaluate whether RS-CL affects language following, we separately measure instruction-following performance, rather than only end-task success, in Fig. 6b, and do not observe any degradation from RS-CL (baseline: 58.3%, RS-CL: 64.5%). In this test, we place two training-seen objects close to each other at the pick-up location and specify which object should be picked and placed at the target location. Because the two objects are placed in close proximity, any rollout that does not clearly select the instructed object is counted as failure.
> \begin{array}{lcc}
> \hline
> \text{Method} & \text{Instruction following} & \text{Full success} \newline
> \hline
> \text{Baseline} & 58.3 & 37.5 \newline
> \text{RS-CL} & 64.5 & 41.6 \newline
> \hline
> \end{array}
>
> Consistent with this, when examining RoboCasa-Kitchen rollouts, we did not find clear cases where RS-CL degraded instruction following. Importantly, RoboCasa-Kitchen inherently requires language-conditioned disambiguation. Scenes often contain multiple candidate objects or targets, and the robot must rely on the instruction to determine the correct action, such as picking an object from the counter and placing it into the sink/stove/cabinet/microwave (or the reverse direction), pressing the correct microwave on/off button when the buttons are spatially adjacent, or selecting the correct left/center/right stove knob to control (please refer to Table 10 for specific tasks in RoboCasa-Kitchen). Across these rollouts, we did not observe evidence that RS-CL harms instruction-following behavior beyond overall task success, and both the baseline and RS-CL generally followed the instruction correctly.
>
> ---
> **[Q2] Failure case analysis**
>
> We find that RS-CL tends to provide smaller gains on tasks where action diversity within each task is relatively low and the baseline representation is already strong (e.g, door/drawer opening/closing tasks, -1.3 to +1.0%). In contrast, RS-CL is more beneficial for tasks with more diverse state-to-action mappings, such as pick-and-place tasks (+4.2 to +11.4%) with varying initial and goal placements, or twisting/insertion tasks (+3.8 to +5.4%) that require more precise state-dependent control. We provide RoboCasa-Kitchen success rates by task group below, where the main numbers are results for 30 demonstrations, with 300-demonstration results in parentheses. We will include this analysis and discussion in the revised manuscript.
> \begin{array}{lccc}
> \hline
> \text{Method} & \text{Pick and Place} & \text{Open and Close} & \text{Others (e.g., twisting knobs or insertion)} \newline
> \hline
> \text{Baseline} & 30.1\ (55.6) & 71.3\ (84.7) & 48.2\ (62.6) \newline
> \text{RS-CL} & 41.5\ (59.8) & 70.0\ (85.7) & 52.0\ (68.0) \newline
> \hline
> \end{array}

---

> > ### Author Rebuttal · Reviewer_LdE5 · 2026-04-03
> >
> > I would prefer to maintain my original scores. While the rebuttal addresses several concerns, some uncertainty remains regarding the dependence on the choice and quality of proprioceptive inputs, and how broadly the method generalizes to noisier or more complex state representations.

---

> > > ### Author Response · Authors · 2026-04-05
> > >
> > > Dear reviewer LdE5,
> > >
> > > Thank you once again for your time and thoughtful efforts in reviewing our paper. We are glad to hear that we addressed most of your initial concerns.
> > >
> > > Regarding the remaining concern on state representation, we would like to emphasize that RS-CL works robustly not only in simulation but also in real-world environments across 10 in-house and generalization tasks (Figure 6). Moreover, RS-CL generalizes beyond a single-arm + gripper setup. We also observe **consistent gains in more complex state representations**, as the reviewer suggested, including GR-1 humanoid manipulation and bimanual Panda arms with two Inspire Hands (38-dimensional state).
> > >
> > > \begin{array}{lc}
> > > \hline
> > > \text{Method} & \text{DexMimicGen - Bimanual panda hands (\\%)} \newline
> > > \hline
> > > \text{Baseline} & 52.7 \newline
> > > \text{RS-CL} & \textbf{57.3} \newline
> > > \hline
> > > \end{array}
> > >
> > > \begin{array}{lc}
> > > \hline
> > > \text{Method} & \text{DexMimicGen - GR-1 humanoid (\\%)} \newline
> > > \hline
> > > \text{Baseline} & 68.9 \newline
> > > \text{RS-CL} & \textbf{73.1} \newline
> > > \hline
> > > \end{array}
> > >
> > > We hope these additional results help address the remaining concern, and thank you again for your valuable feedback and consideration.
> > >
> > > Sincerely, \
> > > Authors

---

### Official Review · Reviewer_QCsh · 2026-03-10

**Soundness:** 2
**Presentation:** 2
**Significance:** 2
**Originality:** 2
**Overall Recommendation:** 4
**Confidence:** 4

**Summary:**

This paper proposes Robot State-aware Contrastive Loss (RS-CL), a representation learning method for Vision-Language-Action (VLA) models. RS-CL uses relative distances between proprioceptive states as soft contrastive supervision to train the VLM representations alongside standard VLA training. The results demonstrate performance improvements across several benchmarks.

**Compliance With Llm Reviewing Policy:**

Affirmed.

**Final Justification:**

Thank you to the authors for the additional clarifications.

I still maintain that using state to align visual information offers a relatively marginal improvement and, as agreed by the authors, its scope of applicability is limited. However, I do acknowledge the visible performance improvements demonstrated in specific scenarios.

Given that the authors are willing to modify their claims and have addressed the majority of my concerns during the rebuttal process, I am raising my score from 3 to 4.

**Key Questions For Authors:**

1. In the analysis of Figure 2b, the claim that the pre-trained VLM representations are “dominated by visual appearance (e.g., distractor objects)” lacks sufficient empirical evidence.

2. Although the robot state is control-relevant, it is still not clear whether aligning the visual representation to this information alone is sufficient. As the authors themselves mention in the conclusion, other signals, such as key object poses, are also very important.

3. Regarding the view-cutoff method, how does the framework handle datasets where only a single camera view is available?

4. For the soft weight calculation on the states, is the method constrained to a single dataset at a time? Different datasets typically have different definitions for the robot state.

5. How is the contrastive signal strength balanced? If the contrastive signal is too strong, the image representation may essentially collapse into the state signal. This could result in the loss of crucial visual information regarding target objects and cause the model to converge to a local optimum.

**Strengths And Weaknesses:**

Strengths:

The proposed method is simple and intuitive. Furthermore, the paper is well-written and easy to follow.

Weaknesses:

Please refer to the questions below.

---

> ### Author Rebuttal · Authors · 2026-03-31
>
> Dear Reviewer QCsh,
>
> We sincerely appreciate your efforts and comments to improve the manuscript. We respond to your comment in what follows.
>
> ---
> **[Q1] Further evidence of visual dominance in VLM representation**
>
> To make this point more explicit, we quantify our claim that pre-trained VLM representations are organized primarily by scene-dependent visual appearance rather than by control-relevant task progress. The task progress is control-relevant because, for identical manipulation tasks, it is closely tied to the robot’s current configuration and next control requirement.  Beyond Fig. 2b, Fig. 14 and Appendix E offer a more detailed visualization where identical task trajectories form groups (G1-G4) according to visual scene characteristics.  We further measured scene classification accuracy on the same embeddings, alongside the task-progress classification in Table 5. The pre-trained VLM representations achieve 99.6% scene classification but only 1.2% task-progress classification, confirming that they are dominated by scene-dependent visual appearance rather than task progress. We also report RS-CL aligned representation, which preserves scene information (93.6%) while improving task-progress structure (22.9%). We refer back to this in our response to [Q5].
> \begin{array}{lcc}
> \hline
> \text{Features} & \text{Scene classification} & \text{Task progress classification} \newline
> \hline
> \text{Pre-trained VLM representations} & 99.6 & 1.2 \newline
> \text{RS-CL aligned representations} & 93.6 & 22.9 \newline
> \text{Ground-truth proprioceptive states} & 53.1 & 25.6 \newline
> \hline
> \end{array}
>
> ---
> **[Q2] Incorporating additional physical signals**
>
> As noted in our conclusion, we do not claim that proprioceptive state is the only control-relevant signal, and other physical cues such as key object poses can also be important. Rather, our intention is to show that shaping VLM representation geometry using physical signals is a promising direction for VLA learning, and we instantiate this idea using proprioceptive state because it is a broadly available signal in standard robot data,  requiring no additional sensors or perception modules. While proprioception alone may not capture every control-relevant factor, our results show that it already provides an effective and practical supervision signal across a broad range of tasks and settings. Extending RS-CL to additional physical signals is a natural and promising direction for future work.
>
> ---
> **[Q3] Single camera view applicability**
>
> We clarify that view-cutoff is not a required component of RS-CL, but an augmentation designed to be more effective in multi-view settings. In single-view settings, RS-CL can instead be combined with other representation-level augmentations, as suggested by Table 4b. We additionally evaluated RS-CL on the single-view GR-1 embodiment of the DexmimicGen benchmark. In this setting, view cutoff is not effective, whereas RS-CL with span cutoff improves performance from 68.9% to 73.1%, showing that RS-CL remains applicable in single-view settings with alternative representation-level augmentations.
> \begin{array}{lc}
> \hline
> \text{Method} & \text{DexmimicGen - Single view GR-1 (\\%)} \newline
> \hline
> \text{Baseline} & 68.9 \newline
> \text{RS-CL (span cutoff)} & 73.1 \newline
> \text{RS-CL (view cutoff)} & 68.7 \newline
> \hline
> \end{array}
>
> ---
> **[Q4] Scope of the soft-weight formulation**
>
> Our soft weight computation assumes a shared state definition, so it is most naturally applied to a single embodiment/dataset at a time. We believe this is still well aligned with the main practical use case of VLA fine-tuning, where data are typically collected for one robot embodiment and the model is adapted to that setting. In this setting, RS-CL provides a simple and effective way to improve downstream task performance. Extending the method to jointly handle multiple datasets with heterogeneous state spaces is an interesting direction for future work, but is beyond the scope of this paper.
>
> ---
> **[Q5] Balance of the contrastive signal**
>
> Our design balances the contrastive signal in two ways. First, RS-CL is trained jointly with the action prediction loss, so it complements visual information with control-relevant structure, rather than replacing it. Empirically, we do not observe degraded generalization with RS-CL (Fig. 6b), and Fig. 14, Appendix E further shows that scene-dependent groups (G1-G4) remain present after alignment (connected by gray lines), rather than collapsing into the state signal. This is also consistent with the analysis in our response to [Q1], where scene information remains highly classifiable after RS-CL. Second, we control the strength of RS-CL using a weighting coefficient λ that decays to 0 with a cosine schedule, so the contrastive objective mainly shapes the representation early in training while later stages focus more on action prediction. This is supported by Table 8, where the cosine schedule outperforms fixed-λ settings.

---

> > ### Author Rebuttal · Reviewer_QCsh · 2026-04-04
> >
> > Thank you for the detailed clarifications. Your response addresses several of my concerns, particularly regarding the single-view setting, the scope of the soft-weight formulation, and the role of the contrastive objective.
> >
> > My remaining reservation is that the new evidence mainly supports scene dependence rather than the stronger claim of distractor dominance, and that the sufficiency of proprioceptive alignment alone is still not fully established.

---

> > > ### Author Response · Authors · 2026-04-05
> > >
> > > Dear Reviewer QCsh,
> > >
> > > Thank you once again for your time and thoughtful efforts in reviewing our paper. We are glad to hear that our rebuttal helped address your concerns regarding the single-view setting, the scope of the soft-weight formulation, and the role of our contrastive objective.
> > >
> > > Regarding the wording of “dominated by visual appearance (e.g., distractor objects)”, our intention was to show that the pre-trained VLM representation is strongly shaped by **visually dominant objects and layouts that are not directly necessary for action prediction**. In Fig. 14, the clusters G1-G4 are associated with large objects in the scene, such as a flat tabletop, stove/burner, or oven, even though these elements are not directly relevant to the action required for the task. We agree that the term “distractor” may be read as referring to smaller task-irrelevant objects, which was not our intended meaning, and we will revise the phrase to make this clearer in the final version.
> > >
> > > On the second point, we would like to clarify again that we do not claim that proprioceptive alignment alone is universally sufficient, as also noted in our conclusion. Rather, our claim is that the proprioceptive state is a **practical and widely available physical signal**, and that **RS-CL can effectively leverage it to provide control-relevant supervision** across diverse tasks and training setups. Extending RS-CL to additional physical signals is an important promising future direction, where we believe our paper is useful for researchers pursuing it.
> > >
> > > We hope this clarification helps address the remaining concerns, and we sincerely appreciate your valuable feedback and consideration.
> > >
> > > Thank you very much, \
> > > Authors

---

### Official Review · Reviewer_u4HZ · 2026-03-12

**Soundness:** 3
**Presentation:** 3
**Significance:** 2
**Originality:** 2
**Overall Recommendation:** 3
**Confidence:** 3

**Summary:**

The paper studies representation learning in Vision-Language-Action (VLA) models, arguing that VLM representations are insufficiently sensitive to robotic signals such as proprioceptive states. Therefore, they introduce Robot State-aware Contrastive Loss (RS-CL), which uses relative distances between robot states as soft supervision in a contrastive objective to align representations with robot proprioception. The method is evaluated on RoboCasa-Kitchen, LIBERO, and real-robot manipulation tasks. Results show consistent improvements in manipulation success rates and achieve state-of-the-art performance on RoboCasa-Kitchen.

**Compliance With Llm Reviewing Policy:**

Affirmed.

**Final Justification:**

Remain the current score.

**Key Questions For Authors:**

Q1: Failure cases of RS-CL. Table 10 shows several tasks in which RS-CL underperforms the baseline (e.g., "Close Single Door" at 30 demos and "Turn Sink Spout" at 300 demos). Do these cases share common patterns, such as task type, contact structure, or data regime? A brief discussion would help clarify when RS-CL may be less effective.

Q2. Choice of proprioceptive state. The paper motivates using proprioceptive-state distance rather than action distance, but does not analyse the state definition itself. Why is the default state mainly based on end-effector pose and gripper state, rather than alternatives such as joint angles? Is this choice equally suitable across tasks with different contact and control structures?

Q3. Need for the summarisation token. RS-CL introduces a learnable summarisation token, but the ablation does not isolate its effect. Could the authors clarify their contribution and whether similar gains can be obtained with a simpler global readout?

**Limitations:**

As noted in Q1, the paper does not sufficiently analyse cases where RS-CL underperforms the baseline. This would help clarify the method's limitations

**Strengths And Weaknesses:**

RS-CL is a simple, technically sound regulariser that fits naturally into standard VLA training. Results show clear gains in manipulation success and robustness. The paper is clearly written and well organised, with enough detail to understand and reproduce the RS-CL module. The contribution is practically relevant to VLM-based manipulation, offering a lightweight way to make existing VLA models more sensitive to the robot's state without changing the backbone.
The paper briefly touches on why RS-CL works via manifold visualisations and a few representation probes, but a deeper analysis of training dynamics and typical failure modes is largely left open. Conceptually, RS-CL mostly refines and combines existing ideas (contrastive regularisation, state-conditioned weighting, view-based augmentation) for VLA, so the originality feels moderate rather than a major paradigm shift.

---

> ### Author Rebuttal · Authors · 2026-03-31
>
> Dear Reviewer u4HZ,
>
> We sincerely appreciate your efforts and comments to improve the manuscript. We respond to your comment in what follows.
>
> ---
> **[W1] Additional analysis and originality of RS-CL**
>
> We appreciate the reviewer’s suggestion for deeper analysis. Here, we provide a more explicit explanation of why state alignment improves action prediction: since the learned representation serves as the conditioning input to the action decoder, organizing it along control-relevant factors makes the mapping from representation to action easier to learn. Please also see our response to Reviewer UWEC [Q1] for additional discussion.
>
> Regarding originality, we highlight a mismatch between generic pre-trained VLM representations and the structure needed for robot control. Our contribution is a principled and practical training framework to address this mismatch through robot-state-aware supervision within single-stage, end-to-end VLA training. We view this as a meaningful contribution to VLA learning because it shows how physical signals such as proprioceptive states can be used to reshape VLM representations toward control-relevant structure in a lightweight and practical way.
>
> ---
> **[Q1] Failure case analysis**
>
> Because RoboCasa-Kitchen is evaluated in a multi-task optimization setting, per-task results can fluctuate due to the training dynamics, and we therefore use the average success rate over all 24 tasks as the primary metric. For example, on the tasks mentioned by the reviewer (“Close Single Door” and “Turn Sink Spout”), RS-CL performs better on the same tasks at 100 demonstrations.
>
> Nevertheless, we find that RS-CL tends to provide smaller gains on tasks where action diversity within each task is relatively low and the baseline representation is already strong (e.g, door/drawer opening/closing tasks, -1.3 to +1.0%). In contrast, RS-CL is more beneficial for tasks with more diverse state-to-action mappings, such as pick-and-place tasks (+4.2 to +11.4%) with varying initial and goal placements, or twisting/insertion tasks (+3.8 to +5.4%) that require more precise state-dependent control. We provide RoboCasa-Kitchen success rates by task group below, where the main numbers are results for 30 demonstrations, with 300-demonstration results in parentheses. We will include this analysis and discussion in the revised manuscript.
> \begin{array}{lccc}
> \hline
> \text{Method} & \text{Pick and Place} & \text{Open and Close} & \text{Others (e.g., twisting knobs or insertion)} \newline
> \hline
> \text{Baseline} & 30.1\ (55.6) & 71.3\ (84.7) & 48.2\ (62.6) \newline
> \text{RS-CL} & 41.5\ (59.8) & 70.0\ (85.7) & 52.0\ (68.0) \newline
> \hline
> \end{array}
>
> ---
> **[Q2] Choice of proprioceptive state**
>
> RS-CL is not tied to a specific proprioceptive state definition. In our main experiments, we use the default state type provided by each VLA framework as the distance signal for RS-CL, where end-effector pose and gripper state are commonly adopted. Importantly, RS-CL remains effective under different state spaces. In the close-lid experiment (Fig. 6b), we use joint positions and still observe gains from RS-CL. To further isolate the effect of the state choice, we compare several definitions while keeping the rest of the framework unchanged, and find consistent improvements over the baseline across all choices.
> \begin{array}{lc}
> \hline
> \text{Method} & \text{RoboCasa-Kitchen (\\%)} \newline
> \hline
> \text{Baseline} & 48.2 \newline
> \text{RS-CL (EEF pose)} & 53.0 \newline
> \text{RS-CL (Joint position)} & 53.4 \newline
> \text{RS-CL (EEF + Joint position)} & 51.5 \newline
> \hline
> \end{array}
> These results suggest that the benefit of RS-CL extends across different state definitions, supporting its applicability under diverse embodiments and task structures. We also refer the reviewer to our responses to [W3] of reviewer UWEC and [Q3] of reviewer QCsh, where we provide additional results on richer state spaces, including bimanual arm with hands and GR-1 humanoid systems.
>
> ---
> **[Q3] Regarding the summarisation token**
>
> To clarify, the core contribution of our work is the robot-state-aware contrastive supervision for VLA representation learning, rather than the summarisation token itself. The learnable token is a practical design choice for applying RS-CL efficiently to long VLM token sequences through a compact global representation, while preserving the full token sequence for action decoding. We additionally compared the learnable summarisation token with global readout methods on RoboCasa-Kitchen, and found that it performs better than using the last token or mean pooling. This suggests that the learnable token provides a more effective dedicated summary pathway for contrastive regularization.
> \begin{array}{lc}
> \hline
> \text{Method} & \text{RoboCasa-Kitchen (\\%)} \newline
> \hline
> \text{Baseline} & 48.2 \newline
> \text{Last token} & 51.1 \newline
> \text{Mean pooling} & 51.8 \newline
> \text{Summarisation token} & 53.0 \newline
> \hline
> \end{array}

---

> > ### Author Rebuttal · Reviewer_u4HZ · 2026-04-03
> >
> > I appreciate the detailed responses. Q1 and Q3 are adequately addressed. About Q2, the combined EEF + Joint condition underperforms both individual definitions, and Joint position outperforms EEF pose, while EEF pose is used as the main setting in the paper. Neither of these points is sufficiently explained.
> > Thanks again for your efforts. I intend to maintain my rating and await the final discussion.

---

> > > ### Author Response · Authors · 2026-04-05
> > >
> > > Dear Reviewer u4HZ,
> > >
> > > Thank you once again for your time and thoughtful efforts in reviewing our paper. We are glad to hear that our rebuttal helped address your concerns regarding Q1 and Q3.
> > >
> > > Regarding your follow-up on Q2, the fact that joint position outperforms EEF pose indicates that our main setting was not specially optimized for performance. In the main paper, we use the **default state representation of the base VLA framework** (GR00T N1.5), i.e., EEF position, 6D rotation, and gripper pose. Our point is that RS-CL already works effectively with the standard state definition used by the original framework. The stronger result with joint positions is also informative, as it suggests that RS-CL may further benefit from a state representation that more completely describes the robot configuration. In particular, joint positions capture the full robot configuration, whereas EEF pose only captures the pose of the end effector.
> > >
> > >
> > > For the combined EEF + Joint condition, we included it to test whether RS-CL remains effective in a larger mixed state space, even though such a combined state representation is not commonly used in VLA models. Since EEF pose and joint position encode overlapping aspects of the same robot configuration, their simple concatenation may introduce **redundancy into the state-distance geometry** rather than making it more informative. This may cause the state distance to overemphasize overlapping aspects  of the robot configuration, rather than better reflect control-relevant similarity. Nevertheless, RS-CL still improves over the baseline in this setting. Importantly, this does not suggest that RS-CL is less effective for more complex or higher-dimensional state spaces. In fact, we observe **consistent gains on more complex embodiments and high-dimensional state spaces**, including GR-1 humanoid manipulation and bimanual Panda arms with two Inspire Hands (38-dimensional state).
> > > \begin{array}{lc}
> > > \hline
> > > \text{Method} & \text{DexMimicGen - Bimanual panda hands (\\%)} \newline
> > > \hline
> > > \text{Baseline} & 52.7 \newline
> > > \text{RS-CL} & \textbf{57.3} \newline
> > > \hline
> > > \end{array}
> > >
> > > \begin{array}{lc}
> > > \hline
> > > \text{Method} & \text{DexMimicGen - GR-1 humanoid (\\%)} \newline
> > > \hline
> > > \text{Baseline} & 68.9 \newline
> > > \text{RS-CL} & \textbf{73.1} \newline
> > > \hline
> > > \end{array}
> > >
> > > We hope this clarification helps address your remaining concern, and we will incorporate this discussion into the revision. Thank you again for your valuable feedback and consideration.
> > >
> > > Sincerely, \
> > > Authors

---

### Official Review · Reviewer_UWEC · 2026-03-23

**Soundness:** 2
**Presentation:** 3
**Significance:** 2
**Originality:** 3
**Overall Recommendation:** 3
**Confidence:** 3

**Summary:**

This paper introduces a representation regularization method of the Robot State-aware Contrastive Loss (RS-CL) for Vision-Language-Action models. It intends to mitigate the representational gap between pre-trained multimodal embeddings and the robotic signals required for precise manipulation. Its motivation is to capture control-relevant cues, such as proprioceptive state and task progress, because existing VLMs trained on Internet-scale vision-language corpora produce representations dominated by visual appearance, especially under end-to-end fine-tuning with an action-prediction objective.

**Compliance With Llm Reviewing Policy:**

Affirmed.

**Key Questions For Authors:**

* Under-explained principle: The paper demonstrates empirical correlation between proprioceptive alignment and policy performance but provides no causal explanation for why such alignment improves action prediction, leaving the core theoretical claim inadequately supported.

* Limited necessary ablations: While Table 4 ablates these designed components separately, there is no experiment that varies them jointly. A 2×2 ablation (with/without soft labels × with/without view cutoff) would clarify whether the two components are additive, synergistic, or redundant, and would provide a cleaner characterization of what drives the gains.

**Limitations:**

Please see the comments in " Strengths And Weaknesses" and "Key Questions for Authors".

**Strengths And Weaknesses:**

# Strengths:

* Clear visual presentation: The work provides a clear visual demonstration that pretrained VLM representations cluster by visual appearance rather than task progress in Figs. 2 and 14, which clearly shows the motivation of the proposed RS-CL. This qualitative evidence is reinforced by quantitative alignment analysis via CKNNA and KNN task-phase classification, shown in Table 5 and Fig. 8.
* Technical soundness: The design of RS-CL is conceptually simple and intuitive. A weighted InfoNCE loss with proprioceptive-distance soft labels is technically sound. Besides, the view cutoff augmentation operates entirely at the representation level, avoiding additional VLM forward passes and adding only ~1.25% wall-clock overhead. The method seamlessly integrates them into existing single-stage VLA training pipelines.
* Comprehensive Ablations: The authors conduct experiments across multiple simulation benchmarks and comparisons with multiple VLM backbones. Ablations are implemented under different training paradigms (fine-tuning and training-from-scratch) and two action modeling paradigms (flow matching and autoregressive). Real-robot experiments further validate the approach's practical applicability, substantially strengthening the empirical case.

# Weakness:
* Unclear technical details: The incorporation of RS-CL computes soft weights based on instantaneous proprioceptive state distances, yet the same state can recur at semantically distinct phases of a trajectory (e.g., approaching vs. retreating from an object). This not only fails to align motion direction or task intent, but also actively pulls together embeddings that the action decoder needs to distinguish.
* Concerns in modular design: Proprioceptively similar states may arise across entirely different tasks (e.g., similar joint configurations during "open microwave" vs. "open cabinet"). RS-CL may treat these as similar and align their VLM embeddings, which is semantically incorrect and may corrupt the representation space.
* Unclear methodology: RS-CL relies exclusively on end-effector position and gripper state as the proprioceptive signal. This is a fairly limited representation of robot state, and it is unclear how the method would perform on tasks that require richer state information (e.g., full joint configurations in highly articulated systems).
* Lack of theoretical analysis: The paper lacks a detailed explanation of why this particular weighting scheme yields better action prediction, as shown in Eq. 4. It remains unclear whether the performance gains stem from the soft weighting or from any proprioceptive-aware contrastive signal. A more rigorous analysis is preferred to strengthen the methodological contribution.

---

> ### Author Rebuttal · Authors · 2026-03-31
>
> Dear Reviewer UWEC,
>
> We sincerely appreciate your efforts and comments to improve the manuscript. We respond to your comment in what follows.
>
> ---
> **[W1, W2] Preservation of semantic information under state-based alignment**
>
> RS-CL is designed to increase control awareness in the representation while preserving the semantic information needed for action prediction. This is because RS-CL is always jointly optimized end-to-end with the action prediction loss. Thus, the representation is not encouraged to collapse by proprioceptive similarity, but instead incorporates state-aware structure while remaining useful for distinguishing the correct action under the context. This is especially important in RoboCasa-Kitchen, where similar initial states require different actions depending on the instruction (please see Table 10 for the specific tasks).
>
> We also verify this in the diverse generalization results in Fig. 6b, and the scene-classification results in our response to Reviewer QCsh [Q1], suggesting that RS-CL preserves semantic structure while improving control-relevant organization.
>
> ---
> **[W3] Performance on richer state information**
>
> We would like to clarify that RS-CL is not restricted to end-effector position and gripper state. On​​ RoboCasa-Kitchen, we also study alternative state definitions, including joint positions and EEF+joint states, and find that they also improve over the baseline.
> \begin{array}{lc}
> \hline
> \text{Method} & \text{RoboCasa-Kitchen (\\%)} \newline
> \hline
> \text{Baseline} & 48.2 \newline
> \text{RS-CL (EEF pose)} & 53.0 \newline
> \text{RS-CL (Joint position)} & 53.4 \newline
> \text{RS-CL (EEF + Joint position)} & 51.5 \newline
> \hline
> \end{array}
>
> We further validate RS-CL on DexMimicGen with bimanual panda arms with Inspire hands (total 38 dims), showing compatibility with higher-dimensional proprioceptive state systems.
> \begin{array}{lc}
> \hline
> \text{Method} & \text{DexMimicGen - Bimanual panda hands (\\%)} \newline
> \hline
> \text{Baseline} & 52.7 \newline
> \text{RS-CL} & 57.3 \newline
> \hline
> \end{array}
>
> ---
> **[W4] Role of the weighting scheme and proprioceptive signal**
>
>
> To isolate the effect, we compare alternatives that use state information without our formulation. As shown in the table below, simply injecting state signals yields only limited or inconsistent improvements, whereas RS-CL achieves a higher success rate. ​​This indicates that the gain depends on how the proprioceptive information is incorporated, not merely on its presence. In this context, the soft weighting in Eq. 4 provides a natural mechanism to incorporate continuous state similarities as supervision. A binary contrastive objective using explicit state-distance threshold is possible, but this requires selecting the threshold and appears less effective in practice.
> \begin{array}{lc}
> \hline
> \text{Method} & \text{RoboCasa-Kitchen (\\%)} \newline
> \hline
> \text{Baseline} & 48.2 \newline
> \text{State in input} & 50.9 \newline
> \text{State reconstruction} & 47.0 \newline
> \text{Binary CL with state threshold} & 50.9 \newline
> \text{RS-CL} & 53.0 \newline
> \hline
> \end{array}
>
> ---
> **[Q1] Why state alignment improves action prediction**
>
> In VLA models, the learned representation serves as the conditioning input to the action decoder, making its structure directly affect action prediction. Consistent with prior work on representation learning for downstream robot control [1,2], we aim to learn representations that better capture control-relevant semantics from observations. However, we find it not well aligned in standard VLM representations (Figs. 2, 14, Table 5). RS-CL addresses this by organizing the representation space according to the robot’s proprioceptive state. As a result, the action decoder can rely on a representation that is already structured along control-relevant factors rather than having to infer it implicitly, making the mapping from representation to action easier to learn.
>
> [1] Sermanet et al., Time-contrastive networks: Self-supervised learning from video, ICRA 2018 \
> [2] Nair et al., R3M: A universal visual representation for robot manipulation, CoRL 2022
>
> ---
> **[Q2] Joint ablation on soft labels and view cutoff**
>
> We conduct a 2x2 ablation on RoboCasa-Kitchen with 300 demos, where the baseline performance is 65.7%. We find that without an appropriate augmentation like view cutoff, the augmented pair remains too similar, and the contrastive objective fails to provide a useful learning signal. Introducing view cutoff makes our contrastive framework effective by creating informative partial-view pairs, and the soft-label supervision further refines the representation by encouraging alignment to robot-state similarity. Thus the soft-label supervision can’t stand alone, but works synergistically with view cutoff.
> \begin{array}{lcc}
> \hline
> \text{} & \text{w/o View cutoff} & \text{View cutoff} \newline
> \hline
> \text{No soft label} & 65.8 & 67.3 \newline
> \text{Soft label} & 65.3 & 69.7 \newline
> \hline
> \end{array}

---

> > ### Author Rebuttal · Reviewer_UWEC · 2026-04-03
> >
> > Thanks for the authors’ response. I am inclined to maintain my current rating.

---

> > > ### Author Response · Authors · 2026-04-05
> > >
> > > Dear reviewer UWEC,
> > >
> > > Thank you once again for your time and thoughtful efforts in reviewing our paper. We are glad to hear we addressed all of your initial concerns.
> > > If you have any further comments or remaining issues, please let us know. In particular, if there is a main concern that we should address for you to consider raising our score, we would value your feedback.
> > >
> > > Sincerely, \
> > > Authors

---

### Official Review · Reviewer_YNCy · 2026-03-24

**Soundness:** 3
**Presentation:** 2
**Significance:** 3
**Originality:** 2
**Overall Recommendation:** 4
**Confidence:** 4

**Summary:**

This paper studies a mismatch in Vision-Language-Action (VLA) learning between pretrained vision-language representations and the needs of robot control. While pretrained VLM features are strong at capturing visual appearance and semantic information, they may not adequately encode control-relevant signals such as robot proprioceptive state. To address this issue, the paper proposes **Robot State-aware Contrastive Loss (RS-CL)**, which uses relative distances between robot states to provide soft supervision for contrastive learning, with the goal of encouraging VLA representations to better reflect control-relevant structure.

The method introduces three main components. First, a **learnable summarisation token** is used to compress long VLM token sequences into a compact representation for contrastive learning. Second, the paper defines a **weighted contrastive objective** in which pairwise similarities are supervised by robot state distances rather than hard labels. Third, the method applies a **view cutoff augmentation** at the representation level to construct augmented views without repeatedly processing raw images through the VLM. RS-CL is trained jointly with the standard action prediction objective in a single-stage pipeline.

The paper evaluates the method on **RoboCasa-Kitchen**, **LIBERO**, and **real-robot tasks**, and also includes experiments under different training settings and model backbones. In addition to task performance, the paper provides ablations and analyses on the contrastive target design, augmentation strategy, and representation quality, aiming to show that robot-state-aware regularisation can improve VLA representations and downstream control performance.

**Compliance With Llm Reviewing Policy:**

Affirmed.

**Final Justification:**

I thank the authors for the detailed rebuttal. The additional experiments and clarifications adequately resolve my main concerns, particularly those regarding the summarisation token, the positioning of the contribution, the scope of applicability, and the presentation of the state-aware weighting details.

Overall, my evaluation remains unchanged. I find the paper technically solid, practically meaningful, and supported by reasonably broad experiments, although I still view the methodological novelty as moderate. On balance, I maintain my original Weak Accept recommendation.

**Key Questions For Authors:**

- Why is a **learnable summarisation token** necessary? Did the authors compare it against mean pooling, CLS pooling, or attention pooling?
- How do the authors position the main novelty of the paper more precisely: as a new contrastive formulation, or as an effective way of injecting robot state information into VLA training?
- What are the actual boundaries of the “general regularizer” claim? Does the method mainly apply to VLA settings with explicit proprioception, multi-view observations, and similar action decoding pipelines?
- Since state-aware supervision is the central contribution, could the paper present the state definition, distance computation, normalisation, and soft-weight construction more clearly in the main text rather than leaving important parts scattered across sections?

**Limitations:**

First, the necessity of the **learnable summarisation token** is not fully established, since the paper does not directly compare it with simpler alternatives such as mean pooling, CLS pooling, or attention pooling. Second, while the use of robot state distance as soft supervision is effective and well motivated, the methodological novelty is moderate from a broader contrastive learning perspective, and the paper could do a better job clarifying what is fundamentally new versus what is primarily a strong task-specific integration. Third, some implementation details related to state definition, normalisation, and soft-weight construction could be presented more clearly in the main text. Finally, although the experiments suggest that the method is robust across several backbones and settings, the broader claim of being a generally plug-in regularizer across VLA families is somewhat stronger than what is fully demonstrated.

**Strengths And Weaknesses:**

## Strengths

**1. The problem is meaningful and well-motivated.**
The paper focuses on a realistic and important issue in VLA: pretrained VLM features are not necessarily well aligned with low-level control requirements. This is a sensible and relevant research question.

**2. The method is lightweight, and the efficiency argument is better supported in the anonymous version.**
RS-CL is introduced as an auxiliary loss within a standard single-stage VLA training pipeline, without requiring a heavy multi-stage design. The paper also includes comparisons to TCN and analyzes FLOPs/runtime overhead, which makes the efficiency claim more convincing.

**3. The empirical evaluation is fairly broad and consistently positive.**
The paper reports improvements on RoboCasa, LIBERO, real-robot tasks, and multiple backbones in from-scratch settings. The real-robot generalisation results also strengthen the empirical case. Overall, the experimental section is reasonably comprehensive and supports the claim that representation regularisation can help VLA performance.

## Weaknesses

**1. The necessity of the summarisation token remains unclear.**
The learnable summarisation token is a reasonable engineering choice for compressing long VLM token sequences, but the paper does not clearly justify why this design is needed over simpler alternatives such as mean pooling, CLS pooling, or attention pooling. Without such an ablation, this component feels more like a practical implementation choice than a validated key innovation.

**2. The methodological novelty is limited.**
At a high level, RS-CL is still a weighted contrastive objective, where the pairwise weights are derived from robot state distance. The paper shows that this design is useful in the VLA setting. However, from a method perspective, I still view the contribution more as a well-motivated task-specific integration than as a substantially new contrastive learning framework.

**3. The “general plug-in regularizer” claim appears somewhat stronger than the current evidence.**
The results across multiple backbones and training settings do suggest a degree of robustness. However, the experiments are still conducted within fairly similar VLA formulations. I think the current evidence supports the claim that the method is helpful across several backbones and setups, but not yet the broader claim that it is generally plug-and-play across different VLA families.

**4. Some key implementation details of the state-aware weighting are still not presented clearly enough in the main paper.**
The paper provides a lot of details, but readers still need to combine information from the method section, implementation details, and appendix to fully reconstruct how state distance, soft weights, and state definitions are handled. Since this is the core of the paper, the exposition could still be clearer and more self-contained.

---

> ### Author Rebuttal · Authors · 2026-03-31
>
> Dear Reviewer YNCy,
>
> We sincerely appreciate your efforts and comments to improve the manuscript. We respond to your comment in what follows.
>
> ---
> **[W1, Q1] Necessity of summarisation token**
>
> We thank the reviewer for the question and conducted an additional comparison with alternative summarisation methods. Our proposed method outperforms alternatives for global feature extraction, such as using the last token [1] or mean pooling on RoboCasa-Kitchen. This suggests that the learnable token is an effective summary mechanism in our setting. Our intuition is that it provides a dedicated, learnable summary pathway for contrastive learning over long VLM token sequences, while remaining separate from the token sequence used by the action decoder. This allows RS-CL to operate on a compact global representation, while the full token sequence continues to be used for action decoding.
> \begin{array}{lc}
> \hline
> \text{Method} & \text{RoboCasa-Kitchen (\\\%)} \newline
> \hline
> \text{Baseline} & 48.2 \newline
> \text{Last token} & 51.1 \newline
> \text{Mean pooling} & 51.8 \newline
> \text{Summarisation token} & 53.0 \newline
> \hline
> \end{array}
> [1] Jiang et al., VLM2Vec: Training Vision-Language Models for Massive Multimodal Embedding Tasks, ICLR 2025
>
> ---
> **[W2, Q2] Positioning of the contribution**
>
> Our main contribution is a principled and practical VLA training framework that shapes VLM representations toward control-relevant structure through robot-state-aware supervision.  We highlight a mismatch between pre-trained VLM representations and the requirements for low-level robot control, and show that RS-CL helps bridge this gap in a lightweight manner. Accordingly, we position RS-CL primarily as an effective robot-state-aware framework for VLA training, rather than as a fundamentally new contrastive learning formulation.
>
> To further support this positioning, we conducted additional experiments comparing different strategies of incorporating robot state into VLA models. While directly providing state as input to the VLM [2] or adding a reconstruction objective yields limited or no gains, RS-CL achieves stronger improvements. This suggests that the benefit comes not merely from injecting state information, but from restructuring the representation space to better reflect control-relevant geometry.
>
> \begin{array}{lc}
> \hline
> \text{Method} & \text{RoboCasa-Kitchen (\\%)} \newline
> \hline
> \text{Baseline} & 48.2 \newline
> \text{State in input} & 50.9 \newline
> \text{State reconstruction} & 47.0 \newline
> \text{RS-CL} & 53.0 \newline
> \hline
> \end{array}
> [2] Pertsch et al., FAST: Efficient Action Tokenization for Vision-Language-Action Models, preprint 2025
>
> ---
> **[W3, Q3] Scope and applicability of RS-CL**
>
> RS-CL is broadly applicable within VLAs that rely on VLM-based representations and explicit robot state signals. In our experiments, we demonstrate consistent improvements across diverse setups, including fine-tuning and from-scratch training (Table 1–2, Fig. 6–7), different VLM backbones (Fig. 7), and varying numbers of observation views (3-view RoboCasa-Kitchen, 2-view LIBERO and real-robot). We further note that RS-CL consistently improves performance across different VLA families with distinct action modeling paradigms (autoregressive vs. flow-matching), as shown in Table 9.
> \begin{array}{lc}
> \hline
> \text{Method} & \text{RoboCasa-Kitchen (\\%)} \newline
> \hline
> \pi_0\text{-FAST (Autoregressive)} & 29.8 \newline
> \pi_0\text{-FAST + RS-CL} & 33.2 \newline
> \text{GR00T-N1.5 (Flow-matching)} & 48.2 \newline
> \text{GR00T-N1.5 + RS-CL} & 53.0 \newline
> \hline
> \end{array}
> At the same time, we would like to clarify the intended scope of the claim. The proposed view-cutoff augmentation is a design choice that further strengthens the framework in multi-view settings by creating informative partial-view pairs for contrastive learning, although alternative augmentations (Table 4b) remain applicable and continue to yield improvements (we also discuss this in our response to [Q3] of Reviewer QCsh, for single-view GR-1 experiments in the Dexmimicgen benchmark). In addition, our current formulation focuses on the proprioceptive state, while incorporating richer physical signals such as object pose or tactile feedback represents a promising direction for future work.
>
> ---
> **[W4, Q4] Regarding presentation of state-aware supervision**
>
> We thank the reviewer for the suggestion. In our main experiments, the state is defined as end-effector position (xyz), 6D rotation, and gripper state (except for the state-choice ablation in real-world experiment), with each dimension min-max normalized to [-1, 1]. We then compute state similarity using L2 distance and convert it into soft weights as in Eq. (4). We will revise the manuscript to make these details clearer in the main method section.

---

> > ### Author Rebuttal · Reviewer_YNCy · 2026-04-01
> >
> > Thank you to the authors for the comprehensive rebuttal and the additional experiments. The authors have adequately addressed all of my initial concerns:
> >
> > 1. Necessity of the summarisation token: The newly provided ablation study comparing the learnable token against 'mean pooling' and 'last token' baselines clearly demonstrates the empirical advantage of your design.
> > 2. Positioning and Novelty: I appreciate the candid clarification regarding the positioning of the paper. Frameing RS-CL as an effective, state-aware framework for VLA rather than a fundamentally new contrastive formulation is accurate. The additional baselines (state in input vs. state reconstruction) further solidify that your specific contrastive integration is highly effective.
> > 3. Scope of the "general regularizer" claim: The supplementary results demonstrating consistent improvements across different action modeling paradigms (autoregressive vs. flow-matching) sufficiently back up your claims regarding the method's versatility. I also appreciate the honest acknowledgment of the current framework's boundaries (e.g., focus on proprioceptive states).
> > 4. Presentation: Thank you for clarifying the specific state definitions, normalization steps, and distance computations, and for committing to moving these crucial implementation details into the main text of the revised manuscript.
> >
> > Based on the strength of this rebuttal, my concerns are fully resolved.

---

> > > ### Author Response · Authors · 2026-04-05
> > >
> > > Dear Reviewer YNCy,
> > >
> > > Thank you once again for your time and thoughtful efforts in reviewing our paper. We are glad to hear we addressed all of your initial concerns.
> > > We appreciate your encouraging assessment, and we will make sure that the revised manuscript reflects these clarifications and additional results as clearly as possible.
> > >
> > > Sincerely, \
> > > Authors

---

### Decision · Program_Chairs · 2026-04-30

**Decision:**

Accept (regular)

**Comment:**

This paper addresses an important problem in vision-language-action learning: pre-trained vision-language representations are often poorly aligned with control-relevant robotic signals. The proposed RS-CL method is a lightweight regularization approach that uses robot-state aware contrastive supervision to reshape representations toward control-relevant structure, while remaining compatible with standard VLA training pipelines.

Reviewers broadly agreed that the work is well motivated, the proposed method is technically reasonable, and the evaluation results are promising. The main weaknesses raised during review concern novelty, scope, and clarity. The rebuttal addressed the majority of the concerns. The remaining weak one is technical novelty.

The AC agrees that this work is technically sound and practically effective. Although the technical novelty is moderate, the idea of relating the VL representation to the robot state is new and will inspire the community to explore further.